# MEG source imaging detects optogenetically-induced activity in cortical and subcortical networks

Gregory E. Alberto [1,5✉], Jennifer R. Stapleton-Kotloski [2,3,5], David C. Klorig [1], Emily R. Rogers[1], Christos Constantinidis[1], James B. Daunais[4] & Dwayne W. Godwin [1,2,3,4]

Magnetoencephalography measures neuromagnetic activity with high temporal, and theoretically, high spatial resolution. We developed an experimental platform combining MEG-compatible optogenetic techniques in nonhuman primates for use as a functional brain-mapping platform. Here we show localization of optogenetically evoked signals to known sources in the superficial arcuate sulcus of cortex and in CA3 of hippocampus at a resolution of 750 $\mu m^3$. We detect activation in subcortical, thalamic, and extended temporal structures, conforming to known anatomical and functional brain networks associated with the respective sites of stimulation. This demonstrates that high-resolution localization of experimentally produced deep sources is possible within an intact brain. This approach is suitable for exploring causal relationships between discrete brain regions through precise optogenetic control and simultaneous whole brain MEG recording with high-resolution magnetic source imaging (MSI).

[1] Wake Forest School of Medicine; Department of Neurobiology and Anatomy, Winston-Salem, NC, USA. [2] Wake Forest School of Medicine Department of Neurology, Winston-Salem, NC, USA. [3] Research and Education Department, W.G. (Bill) Hefner Veterans Affairs Medical Center, Salisbury, NC, USA. [4] Wake Forest School of Medicine Department of Physiology and Pharmacology, Winston-Salem, NC, USA. [5]These authors contributed equally: Gregory E. Alberto, Jennifer R. Stapleton-Kotloski. ✉email: gregory.e.alberto@hitchcock.org

The ultimate goal of systems neuroscience is to be able to understand, interpret, and predict the distributed patterns of measurable physiologic activity that underpin neural computations and give rise to behavior. To this end, capturing the causal relationships of the brain's complex subsystems at the temporal and spatial scales across which computation occurs is particularly important. Whole-brain functional neuroimaging techniques like functional magnetic resonance imaging (fMRI) and positron emission tomography (PET) offer the ability to observe such distributed relationships, but with qualifying limitations. For example, fMRI and PET generally possess excellent spatial resolution but limited temporal resolution relative to neuron population activity. Alternatively, MEG and electroencephalography (EEG) have excellent temporal resolution but are often thought to have a poor spatial resolution.

MEG holds particular promise for providing high resolution in both the spatial and temporal domains. As with EEG, MEG has sub-millisecond temporal resolution but with the added potential for superior spatial resolution[1,2]. The biomagnetic signals detected by MEG are not distorted by the intervening tissues of the head, unlike the electric fields measured by EEG, thus allowing for a more precise estimate of the locations of these signals with MSI than is feasible with EEG[2] source localization.

Uncertainty remains over the ability to detect deep brain activity with MEG. This uncertainty stems, in part, from early studies employing simultaneous intracranial EEG and MEG in patients with epilepsy that suggested MEG was unable to detect activity originating in deep structures such as the hippocampus[3–5]. However, the specific MEG hardware utilized, analytic techniques applied, and even the recording environment all contribute to MSI accuracy. The signal-to-noise ratio can be improved through the use of axial gradiometers (shown to have a superior depth profile in comparison to planar gradiometers[6,7]), a low noise floor, minimal motion[8,9], and appropriate use of beamformers[10–12]. In theory, attention to these parameters should allow for sub-mm$^3$ localization of activity[8] throughout the volume of the brain. This theoretical resolution is supported by the use of MEG in the pre-surgical evaluation of epilepsy[11,13–15] and in studies of deep structures such as hippocampus[12] and amygdala[16], though direct evidence of detection of a known source is lacking.

We have developed a nonhuman primate (NHP) optogenetic model for use in MEG with the goal of measuring whole-brain responses to experimenter-controlled optogenetic stimulation. Successful implementation of this combination of techniques provides a platform for functional brain mapping. Importantly, optogenetic methods are magnetically silent and allow for precise control over neural responses through the activation of virally expressed, light-sensitive proteins[17], without obscuring biomagnetic signals.

Here we report the ability to selectively activate and accurately localize neuronal activity to the site of stimulation in two representative brain regions—the arcuate sulcus and hippocampus—at sub-mm$^3$ resolution. Furthermore, we demonstrate the ability to localize discrete sources of activity in brain regions downstream from stimulated areas that are consistent with known functional and anatomical networks. These findings demonstrate the utility of combining optogenetics and MEG as a platform for experimenter-controlled functional brain mapping across a wide range of spatiotemporal scales.

In this report, we have utilized synthetic aperture magnetometry (SAM)[10], a linearly constrained minimum variance (LCMV) beamformer method[10–12] with a theoretical sub-mm$^3$ spatial resolution[6,8,9]. We have used SAM to convert MEG signals into whole-brain statistical parametric maps (SPMs), differentiating it from more traditional EEG and MEG dipole analyses[10]. In doing this we also demonstrate accurate detection of deep activity to a known source using SAM, adding empirical support to the existing body of theoretical evidence for deep signal detection and the potentially high resolution of MSI. We propose that the tight temporal and spatial control over neural populations afforded with optogenetics coupled with whole brain, high spatiotemporal resolution MEG recordings will advance efforts in functional neuroimaging.

## Results

**Electrophysiology of early expression.** Three vervet monkeys (Chlorocebus aethiops; referred to as M1, M2, and M3) received intracerebral injections of AAV2/10-CaMKIIa-ChR2-eYFP and were implanted with depth electrodes and optical fibers. We found the AAV2/10 to be an effective serotype for transducing neurons in NHP in pilot experiments. Optogenetically evoked activity measured via local field potential (LFP) recordings was used to track the expression of ChR2 at the implantation site. All reported experimentation and data acquisition occurred under anesthesia. Functional expression was determined using optogenetic intensity-response curves at each site 2-, 5-, and 7-week post-injection. An optogenetically evoked potential was detectable from depth recordings by 5 weeks post-expression and had stabilized by ~7 weeks (Fig. 1c–e; cortical: baseline (2 week) 95% confidence interval (CI) in mV [−0.151, 0.374], post-injection (5 weeks) 95% CI [0.522, 1.12], post-expression (7 weeks) 95% CI [0.683, 1.09], hippocampal: baseline (2 weeks) 95% CI [−4.55e−3, 0.455], post-expression (5 weeks) 95% CI [1.31, 1.65], post-expression (7 weeks) 95% CI [1.50, 1.89]).

**MEG recordings of early expression.** We analyzed post-surgical (2 weeks) MEG data using SAM. Consistent with the electrophysiological recordings obtained in the same week, there were no detectable changes (local maxima or minima) in the SAM SPMs at the site of stimulation, and the maps were qualitatively similar to the pre-surgical control scans (Not shown—see "Methods" for general beamforming parameters). We conclude that expression of ChR2 at two weeks post-injection was insufficient to generate an optically evoked response that could be identified using SAM, which is further evidence that our preparation is insensitive to light stimulation in the absence of ChR2.

Postmortem, fluorescence confocal microscopy revealed ChR2-eYFP at the targeted sites of transduction, indicating uptake of the vector by the surrounding neurons and expression of the transgene. Labeled neurons were observed in the posterior wall of arcuate sulcus corresponding to area 8a of the monkey cortex[18–20] (Fig. 1ai). Hippocampal imaging revealed ChR2-eYFP labeled neurons in anterior dorsal hippocampus corresponding to area CA3[19] with labeled efferent connections observable in the surrounding tissue (Fig. 1bi). Structural MRI prior to necropsy confirmed placement of the electrode coupled optical fiber (optrode) in both the posterior wall of arcuate sulcus and in anterior dorsal hippocampus[20] (Fig. 1aii–iii and Fig. 1bii–iii).

As a positive control for SAM detection and localization accuracy, we presented pulses of white light to the left lower quadrant of the visual field of the left eye of NHP M1 (the right eye was occluded). Peaks in the SAM SPMs (expressed as pseudo-t scores and represented as green and yellow points on the associated MRI slices—see ONLINE METHODS, Analysis for more details) indicate significant voxels (sources)[2,5] in comparison to baseline and represent voxels of highly synchronous (red) or desynchronous (blue) activity within the frequency band of interest. All brain images are depicted in radiological coordinates, in which the right hemisphere is presented on the left.

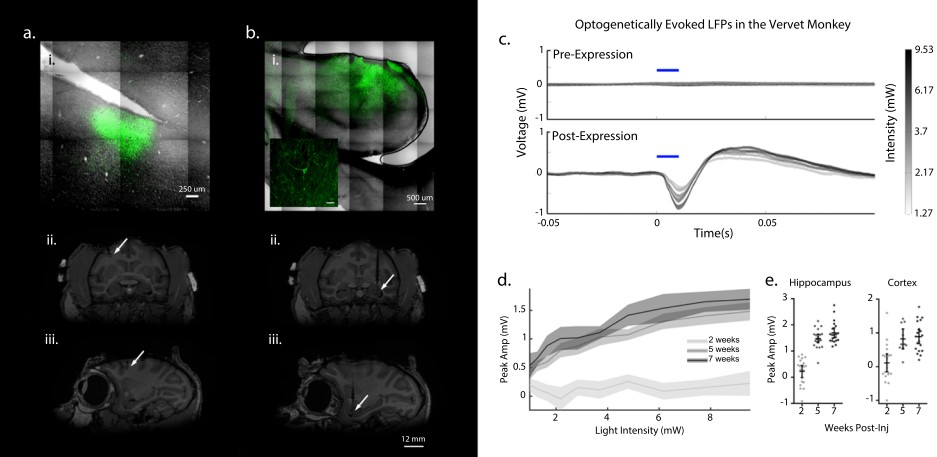

**Fig. 1 Location of optogenetic protein expression and light evoked responses. a** Cortical transduction and implantation. **i** ChR2-eYFP labeled neurons confined to gray matter of the posterior wall of right arcuate sulcus corresponding to area 8a. Coronal (**ii**) and sagittal (**iii**) MRI demonstrating location of the optrode in terminal MRI (arrow). **b** Hippocampal transduction and implantation. **i** ChR2-eYFP labeled neurons in the anterior dorsolateral aspect of left hippocampus corresponding to area CA3 indicate transgene expression; inset demonstrating labeled neuron. Coronal (**ii**) and sagittal (**iii**) MRI demonstrating location of the optrode in terminal MRI (arrow). **c** Example of averaged response to stimulation at five intensity levels (gray to black) in hippocampus showing absence of response in week 2 (upper trace) compared to the large, biphasic LFP in week 7 (lower trace). Blue bar indicates stimulus onset and duration. **d** Average intensity-response curves for each of the three-time points in representative animal M1.Bands = 95% CI. **e** Maximum intensity stimulation (9.53 mW) of cortex and hippocampus at 7 weeks post-injection elicited a response as measured by the LFP electrode that was greater than at 2 weeks post-injection in representative animal M1. ($n = 20$ trials, except $n = 10$ trials for cortex at 5 weeks). Center line = mean, error bars = 95% CI.

In contrast to the lack of optogenetic biomagnetic responses at 2 weeks, visual stimulation experiments conducted at this time point demonstrated SAM peak voxels in the right primary visual cortex/occipital lobe (±50 ms map window relative to stimulus onset, bandwidth of 20–35 Hz, pseudo-t's = 0.3) consistent with the early cortical component of visual evoked potentials[21] (Table 1; Fig. S1aix, xi). In addition to activation in occipital cortex, SAM also revealed cortical and *subcortical* peaks associated with the visual network, including areas corresponding to left optic tract, left lateral geniculate n., and left superior colliculus. See Table 1 (Visual) for details of anatomic peak locations and for the associated SAM parameters. Importantly, arcuate and hippocampal responses were also detected in the LFP in response to visual stimulation and were similarly detected in the SAM map (Fig. S1b).

MSI can also be used to construct virtual electrodes (source series) for any voxel in the brain, providing a continuous, wide-band, sub-millisecond readout of activity that provides similar neurophysiological data relative to the LFP recorded from an actual invasive electrode[11,13,22–24]. Here, the averaged visual evoked field (Fig. S1c) was extracted from the virtual electrode derived from right V1 (Fig. S1axi) and shows approximate peak components that compare well with the evoked response (LFP)[24] in Fig. S1b and correspond to the M50 peak. The visual stimulation experiment, in addition to serving as a positive control for evoked signals, also demonstrates that SAM can localize primary sensory cortex in NHPs in response to stimulation, that it can reveal subcortical brain structures that are involved in the processing of visual stimulation, and that activated white matter tracts may also be localized with SAM.

**MEG recordings of stable expression**. Eight weeks after transduction, and once the neural response to optical stimulation had reached a plateau, MEG recordings commenced. Figure 2a depicts an example of optogenetic activation (50 ms square pulses) in M1 as localized by dual-state SAM to the transduction site in the posterior

wall of the right arcuate sulcus in the coronal (i), sagittal (ii), and axial planes (iii). The whole brain, un-thresholded, dual-state SAM SPM is depicted in Fig. 2aiv; note the region of synchronized activity (red voxels) arising from the site of activation and surrounded by desynchronized activity (blue voxels). See Table 1 (Cortex) for additional peak localization and analysis parameters.

Figure 2b depicts the SAM peaks within the arcuate sulcus corresponding to four additional and different optogenetic stimulus inputs in two different animals, M1 and M2 (See Table 1 for animal ID, stimulus type, and analysis details). Figure 2bv presents a single trial of a local field potential (white) recorded from the indwelling optrode and a simultaneous SAM virtual electrode (red, viewing filters set from 1 to 30 Hz) extracted from the arcuate peak shown in Fig. 2biii for M2.

Consistent activity was also elicited in the left hippocampi of M2 and M3 in response to a variety of optical stimulus types. These variable stimulus types were localized to the known hippocampal source across animals indicating that SAM localization is robust to a combination of optical stimulus types, individual anatomy, and recording conditions across subjects (See Table 1 for subject ID, stimulus details and analysis parameters). Figure 3a presents an example of the left hippocampal peak co-localized to the site of optical stimulation in M3. Figure 3a depicts the peak in the coronal (i), sagittal (ii), and axial planes (iii), and Fig. 3aiv illustrates the un-thresholded, whole-brain SAM dual-state map associated with the 20 Hz pulse input. Note the narrow band of synchronization (red) confined to the dorsal aspect of the left hippocampus, and synchronous activity propagating towards left insula. The second row of Fig. 3 depicts the left hippocampal peaks elicited by additional optogenetic stimulation for two different NHPs, M1 and M3, again demonstrating a consistent ability to detect deep signals across subjects and stimulus types. Figure 3bv presents a single trial of a local field potential and the simultaneous SAM virtual electrode extracted from the corresponding hippocampal peak. As with the arcuate source series, both the LFP and virtual electrode qualitatively share similar features and time courses on a single trial basis.

**Table 1 Experimental details and results.**

| Study | Stim type | Stim pattern | Stim power (mW) | Pulse duration (ms) | Train duration (s) | ISI (s) | Duty cycle (%) | Fiber diameter (μm) | Stim location | Monkey ID | Laterality | Anatomic location of peak | Analysis type | Window | Bandwidth | Voxel Size (mm) | Pseudo-t score | Figure |
|---|---|---|---|---|---|---|---|---|---|---|---|---|---|---|---|---|---|---|
| Visual | Visual | 10 ms pulse | n.a. | 10 | | 6 | n.a. | n.a. | Left retina | M1 | Left | Optic tract | Dual state | ±75 ms | 12–35 Hz | 0.75 | 0.4 | S1aiii |
| | Visual | 10 ms pulse | n.a. | 10 | | 6 | n.a. | n.a. | Left retina | M1 | Left | Lateral geniculate | Dual state | ±50 ms | 20–50 Hz | 0.75 | 0.4 | S1aiv |
| | Visual | 10 ms pulse | n.a. | 10 | | 6 | n.a. | n.a. | Left retina | M1 | Left | Superior colliculus | Dual state | ±50 ms | 20–60 Hz | 0.75 | 0.4 | S1avii |
| | Visual | 10 ms pulse | n.a. | 10 | | 6 | n.a. | n.a. | Left retina | M1 | Right | Arcuate cortex | Dual state | ±75 ms | 12–120 Hz | 0.75 | 0.6 | S1ai |
| | Visual | 10 ms pulse | n.a. | 10 | | 6 | n.a. | n.a. | Left retina | M1 | Left | Anterior commissure | Dual state | ±75 ms | 12–35 Hz | 0.75 | 0.6 | S1aii |
| | Visual | 10 ms pulse | n.a. | 10 | | 6 | n.a. | n.a. | Left retina | M1 | Right | Posterior hippocampus | Dual state | ±75 ms | 12–55 Hz | 0.75 | 0.3 | S1av |
| | Visual | 10 ms pulse | n.a. | 10 | | 6 | n.a. | n.a. | Left retina | M1 | Right | Posterior white matter tracts | Dual state | ±75 ms | 12–60 Hz | 0.75 | 0.4 | S1avi |
| | Visual | 10 ms pulse | n.a. | 10 | | 6 | n.a. | n.a. | Left retina | M1 | left | Secondary visual cortex, V2 | Dual state | ±75 ms | 12–65 Hz | 0.75 | 0.4 | S1aviii |
| | Visual | 10 ms pulse | n.a. | 10 | | 6 | n.a. | n.a. | Left retina | M1 | left | Dorsal aspects of V1 | Dual state | ±50 ms ms | 20–45 Hz | 0.75 | 0.7 | S1ax |
| | Visual | 10 ms pulse | n.a. | 10 | | 6 | n.a. | n.a. | Left retina | M1 | right | Dorsal aspects of V1 | Dual state | ±50 ms | 20–45 Hz | 0.75 | 0.3 | S1ax |
| | Visual | 10 ms pulse | n.a. | 10 | | 6 | n.a. | n.a. | Left retina | M1 | left | Primary visual cortex, V1 | Dual state | ±75 ms | 12–65 Hz | 0.75 | 0.4 | S1axii |
| Voxel size | Visual | 10 ms pulse | n.a. | 10 | | 6 | n.a. | n.a. | Left Retina | M1 | left | Occipital cortex | Dual state | ±50 ms | 20–45 Hz | 0.75 | 0.5 | S1axiii |
| | Opto | 50 ms square pulse | 9.53 | 50 | | 6 | 1 | 400 | Right arcuate cortex | M1 | Right | Arcuate peak | Dual state | ±75 ms | 15–35 Hz | 0.75 | 0.6 | S2a |
| | Opto | 50 ms square pulse | 9.53 | 50 | | 6 | 1 | 400 | Right arcuate cortex | M1 | Right | Arcuate | Dual state | ±75 ms | 15–35 Hz | 1 | 0.6 | S2b |
| | Opto | 50 ms square pulse | 9.53 | 50 | | 6 | 1 | 400 | Right arcuate cortex | M1 | Right | Arcuate | Dual state | ±75 ms | 15–35 Hz | 1.5 | 0.5 | S2c |
| | Opto | 50 ms square pulse | 9.53 | 50 | | 6 | 1 | 400 | Right arcuate cortex | M1 | | Unable to detect peak | Dual state | ±75 ms | 15–35 Hz | 2 | N/A | S2d |
| Cortical stimulation | Opto | 50 ms square pulse | 9.53 | 50 | | 6 | 1 | 400 | Right arcuate cortex | M1 | Right | Arcuate peak | Dual state | ±75 ms | 15–35 Hz | 0.75 | 0.6 | 2ai–iv |
| | Opto | 8 Hz sine wave | 8.09 | | 3 | 6 | 50 | 400 | Right arcuate cortex | M1 | Right | Arcuate peak | Dual state | ±3.5 s | DC–20 Hz | 0.75 | 0.5 | 2bi |
| | Opto | 40 Hz sine wave | 8.09 | | 3 | 6 | 50 | 400 | Right arcuate cortex | M1 | Right | Arcuate peak | Dual state | ±3.5 s | 20–50 Hz | 0.75 | 0.7 | 2bii |
| | Opto | 10 ms square pulse | 28 | 10 | 0.5 | 6 | 0.17 | 200 | Right arcuate cortex | M2 | Right | Arcuate peak | Dual state | ±600 ms | 3–70 Hz | 0.75 | 0.2 | 2biii |
| | Opto | 20 Hz pulse train | 28 | 5 | | 6 | 10 | 200 | Right arcuate cortex | M2 | Right | Arcuate peak | Dual state | ±200 ms | 5–70 Hz | 0.75 | 0.3 | 2biv |
| | Opto | 10 ms square pulse | 28 | 10 | | 6 | 0.17 | 200 | Right arcuate cortex | M2 | Right | Virtual electrode | Dual state | | 1–30 Hz | | | 2bv |
| Hippocampal stimulation | Opto | 20 Hz square pulse | 9.53 | 4 | 60 | | 8 | 400 | CA3 of left hippocampus | M3 | Left | Hippocampus, CA3 | Dual state | ±4 s | 10–30 Hz | 0.75 | 1.7 | 3a |
| | Opto | pulse train ramp pulse | 9.53 | 150 | | 6 | <2.5 | 400 | CA3 of left hippocampus | M3 | Left | Hippocampus, CA3 | Dual state | ±150 ms | 8–45 Hz | 0.75 | 0.8 | 3bi |
| | Opto | 10 ms square | 9.53 | 10 | | 6 | 0.17 | 400 | CA3 of left hippocampus | M3 | Left | Hippocampus, CA3 | Dual state | ±150 ms | 7–85 Hz | 0.75 | 0.2 | 3bii |
| | Opto | 8 Hz sine wave | 8.09 | | 3 | 6 | 50 | 400 | CA3 of left hippocampus | M1 | Left | Hippocampus, CA3 | Dual state | ±2 s | DC–15 Hz | 0.75 | 1 | 3biii–iv |
| | Opto | 40 Hz sine wave | 8.09 | | 3 | 6 | 50 | 400 | CA3 of left hippocampus | M1 | Left | Hippocampus, CA3 | Dual state | ±2 s | 20–55 Hz | 0.75 | 0.6 | 3biv |
| | Opto | ramp pulse | 9.53 | 150 | | 6 | <2.5 | 400 | CA3 of left hippocampus | M3 | Left | Virtual electrode | Dual state | ±3 s | 1–35 Hz | 0.75 | | 3bv |
| Mapping CSPTC | Opto | 8 Hz sine wave | 8.09 | | 3 | 6 | 50 | 400 | Right arcuate cortex | M1 | Right | Anterior bank of the arcuate sulcus | Dual state | ±1 s | DC–12 Hz | 0.75 | 0.3 | 4ai |
| | Opto | 8 Hz sine wave | 8.09 | | 3 | 6 | 50 | 400 | Right arcuate cortex | M1 | Right | Corona radiata/dorsolateral tip of the caudate n. | Dual state | ±4 s | DC–15 Hz | 0.75 | 0.5 | 4aiii |
| | Opto | 8 Hz sine wave | 8.09 | | 3 | 6 | 50 | 400 | Right arcuate cortex | M1 | Right | Caudate n. | Dual state | ±2.5 s | DC–50 Hz | 0.75 | 0.7 | 4aiv |
| | Opto | 8 Hz sine wave | 8.09 | | 3 | 6 | 50 | 400 | Right arcuate cortex | M1 | Right | Putamen/external segment of globus pallidus | Dual state | ±4 s | DC–15 Hz | 0.75 | 0.5 | 4av |
| | Opto | 8 Hz sine wave | 8.09 | | 3 | 6 | 50 | 400 | Right arcuate cortex | M1 | Left | Posterior bank of contralateral (left) arcuate sulcus | Dual state | ±3 s | DC–15 Hz | 0.75 | 0.4 | 4avi/vii |
| | Opto | 8 Hz sine wave | 8.09 | | 3 | 6 | 50 | 400 | Right arcuate cortex | M1 | Right | Globus pallidus | Dual state | ±4 s | DC–15 Hz | 0.75 | 0.4 | 4aviii |
| | Opto | 8 Hz sine wave | 8.09 | | 3 | 6 | 50 | 400 | Right arcuate cortex | M1 | Right | Ventroposterior n. of thalamus | Dual state | ±4 s | DC–15 Hz | 0.75 | 0.4 | 4aix |
| | Opto | 8 Hz sine wave | 8.09 | | 3 | 6 | 50 | 400 | Right arcuate cortex | M1 | Right | Medial dorsal n. of thalamus | Dual state | ±4 s | DC–15 Hz | 0.75 | 0.4 | 4ax |
| | Opto | 8 Hz sine wave | 8.09 | | 3 | 6 | 50 | 400 | Right arcuate cortex | M1 | Right | Ventral posterior thalamus | Dual state | ±3.5 s | DC–20 Hz | 0.75 | 0.3 | 4axi |

## Table 1 (continued)

| Study | Stim type | Stim pattern | Stim power (mW) | Pulse duration (ms) | Train duration (s) | ISI (s) | Duty cycle (%) | Fiber diameter (µm) | Stim location | Monkey ID | Laterality | Anatomic location of peak | Analysis type | Window | Bandwidth | Voxel Size (mm) | Pseudo-t score | Figure |
|---|---|---|---|---|---|---|---|---|---|---|---|---|---|---|---|---|---|---|
| | Opto | 8 Hz sine wave | 8.09 | | 3 | 6 | 50 | 400 | Right arcuate cortex | M1 | Right | Parietal-occipital association area of the intraparietal sulcus | Dual state | ±2 s | DC–50 Hz | 0.75 | 0.6 | 4axii |
| | Opto | 20 Hz pulse train | 28 | 5 | 0.5 | 6 | 10 | 200 | Right arcuate cortex | M2 | Right | Caudate n. | Dual state | ±200 ms | 5–70 Hz | 0.75 | 0.3 | 4bi |
| | Opto | 20 Hz pulse train | 28 | 5 | 0.5 | 6 | 10 | 200 | Right arcuate cortex | M2 | Right | Corona radiata | Dual state | ±200 ms | 5–70 Hz | 0.75 | 0.3 | 4bi |
| | Opto | 20 Hz pulse train | 28 | 5 | 0.5 | 6 | 10 | 200 | Right arcuate cortex | M2 | Right | Thalamus | Dual state | ±200 ms | 5–70 Hz | 0.75 | 0.3 | 4bii |
| | Opto | 20 Hz pulse train | 28 | 5 | 0.5 | 6 | 10 | 200 | Right arcuate cortex | M2 | Right | Caudate n. | Dual state | ±200 ms | 5–70 Hz | 0.75 | 0.3 | 4bii |
| | Opto | 20 Hz pulse train | 28 | 5 | 0.5 | 6 | 10 | 200 | Right arcuate cortex | M2 | Right | Posterior dorsal bank of the arcuate sulcus | Dual state | ±200 ms | 5–70 Hz | 0.75 | 0.3 | 4biii |
| | Opto | 20 Hz pulse train | 28 | 5 | 0.5 | 6 | 10 | 200 | Right arcuate cortex | M2 | Right | Thalamus | Dual state | ±200 ms | 5–70 Hz | 0.75 | 0.3 | 4biii |
| | Opto | 20 Hz pulse train | 28 | 5 | 0.5 | 6 | 10 | 200 | Right arcuate cortex | M2 | Right | Thalamus | Dual state | ±200 ms | 5–70 Hz | 0.75 | 0.3 | 4biv |
| | Opto | 20 Hz pulse train | 28 | 5 | 0.5 | 6 | 10 | 200 | Right arcuate cortex | M2 | Right | Posterior ventral bank of arcuate sulcus | Dual state | ±200 ms | 5–70 Hz | 0.75 | 0.3 | 4bv |
| | Opto | 20 Hz pulse train | 28 | 5 | 0.5 | 6 | 10 | 200 | Right arcuate cortex | M2 | Right | Corpus callosum | Dual state | ±200 ms | 5–70 Hz | 0.75 | 0.3 | 4bv |
| | Opto | 20 Hz pulse train | 28 | 5 | 0.5 | 6 | 10 | 200 | Right arcuate cortex | M2 | Right | Caudate n. | Dual state | ±200 ms | 5–70 Hz | 0.75 | 0.3 | 4bvi |
| | Opto | 20 Hz pulse train | 28 | 5 | 0.5 | 6 | 10 | 200 | Right arcuate cortex | M2 | Right | Caudate n. | Dual state | ±200 ms | 5–70 Hz | 0.75 | 0.3 | 4bvi |
| | Opto | 20 Hz pulse train | 28 | 5 | 0.5 | 6 | 10 | 200 | Right arcuate cortex | M2 | Right | Mesencephalic nuclei | Dual state | ±200 ms | 5–70 Hz | 0.75 | 0.3 | 4bvii |
| | Opto | 20 Hz pulse train | 28 | 5 | 0.5 | 6 | 10 | 200 | Right arcuate cortex | M2 | Right | Deep mesencephalic n. | Dual state | ±200 ms | 5–70 Hz | 0.75 | 0.3 | 4bvii |
| | Opto | 20 Hz pulse train | 28 | 5 | 0.5 | 6 | 10 | 200 | Right arcuate cortex | M2 | Right | Substantia nigra | Dual state | ±200 ms | 5–70 Hz | 0.75 | 0.3 | 4bviii |
| | Opto | 20 Hz pulse train | 28 | 5 | 0.5 | 6 | 10 | 200 | Right arcuate cortex | M2 | Right | Tegmental n. | Dual state | ±200 ms | 5–70 Hz | 0.75 | 0.3 | 4bix |
| | Opto | 20 Hz pulse train | 28 | 5 | 0.5 | 6 | 10 | 200 | Right arcuate cortex | M2 | Right | Pontine or mesencephalic n. | Dual state | ±200 ms | 5–70 Hz | 0.75 | 0.3 | 4bx |
| | Opto | 20 Hz pulse train | 28 | 5 | 0.5 | 6 | 10 | 200 | Right arcuate cortex | M2 | rRight | Primary motor cortex | Dual state | ±300 ms | 4–70 Hz | 0.75 | 0.2 | 4bxi/xii 5ai |
| Hippocampal mapping | Opto | 20 Hz pulse train | 9.53 | 4 | 60 | | 8 | 400 | CA3 of left hippocampus | M3 | Left | Insular proisocortex/temporopolar proisocortex | Dual state | ±4 s | 10–30 Hz | 0.75 | 2.3 | 5ai |
| | Opto | 20 Hz pulse train | 9.53 | 4 | 60 | | 8 | 400 | CA3 of left hippocampus | M3 | Left | Pra-insular cortex | Dual state | ±4 s | 10–30 Hz | 0.75 | 2.4 | 5aii |
| | Opto | 20 Hz pulse train | 9.53 | 4 | 60 | | 8 | 400 | CA3 of left hippocampus | M3 | Left | Dorsolateral amygdala | Dual state | ±4 s | 10–30 Hz | 0.75 | 2 | 5aiii |
| | Opto | 20 Hz pulse train | 9.53 | 4 | 60 | | 8 | 400 | CA3 of left hippocampus | M3 | Left | Bsolateral/lateral amygdala | Dual state | ±4 s | 10–30 Hz | 0.75 | 2 | 5aiv |
| | Opto | 20 Hz pulse train | 9.53 | 4 | 60 | | 8 | 400 | CA3 of left hippocampus | M3 | Left | Hippocampus* | Dual state | ±4 s | 10–30 Hz | 0.75 | 1.7 | 5av |
| | Opto | 20 Hz pulse train | 9.53 | 4 | 60 | | 8 | 400 | CA3 of left hippocampus | M3 | Left | Parasubiculum/presubiculum of hippocampus | Dual state | ±4 s | 10–30 Hz | 0.75 | 1.6 | 5avi |
| | Opto | 20 Hz pulse train | 9.53 | 4 | 60 | | 8 | 400 | CA3 of left hippocampus | M3 | Right | Hippocampus | Dual state | ±4 s | 10–30 Hz | 0.75 | 2.6 | 5avii |
| | Opto | 20 Hz pulse train | 9.53 | 4 | 60 | | 8 | 400 | CA3 of left hippocampus | M3 | Left | Pontine reticular activating formation | Dual state | ±4 s | 10–30 Hz | 0.75 | 2 | 5aviii |
| | Opto | 8 Hz sine wave | 8.09 | | 3 | 6 | 50 | 400 | CA3 of left hippocampus | M1 | Left | Insula | Dual state | ±2 s | DC–15 Hz | 0.75 | 1 | 5bi |
| | Opto | 8 Hz sine wave | 8.09 | | 3 | 6 | 50 | 400 | CA3 of left hippocampus | M1 | Left | PGa/IPa association areas of temporal cortex | Dual state | ±2 s | DC–15 Hz | 0.75 | 1 | 5bii |
| | Opto | 8 Hz sine wave | 8.09 | | 3 | 6 | 50 | 400 | CA3 of left hippocampus | M1 | Left | Hippocampus | Dual state | ±2 s | DC–15 Hz | 0.75 | 1 | 5bii |
| | Opto | 8 Hz sine wave | 8.09 | | 3 | 6 | 50 | 400 | CA3 of left hippocampus | M1 | Left | Red n. | Dual state | ±2 s | DC–15 Hz | 0.75 | 0.6 | 5biii |
| | Opto | 8 Hz sine wave | 8.09 | | 3 | 6 | 50 | 400 | CA3 of left hippocampus | M1 | Left | Area TEA/TEM of temporal cortex | Dual state | ±2 s | DC–15 Hz | 0.75 | 1 | 5biii |
| | Opto | 8 Hz sine wave | 8.09 | | 3 | 6 | 50 | 400 | CA3 of left hippocampus | M1 | Right | Area TL/TFM of right temporal cortex | Dual state | ±2 s | DC–15 Hz | 0.75 | 0.9 | 5biv |

**Table 1 (continued)**

| Study | Stim type | Stim pattern | Stim power (mW) | Pulse duration (ms) | Train duration (s) | ISI (s) | Duty cycle (%) | Fiber diameter (μm) | Stim location | Monkey ID | Laterality | Anatomic location of peak | Analysis type | Window | Bandwidth | Voxel Size (mm) | Pseudo-t score | Figure |
|---|---|---|---|---|---|---|---|---|---|---|---|---|---|---|---|---|---|---|
| | Opto | 40 Hz sine wave | 8.09 | | 3 | 6 | 50 | 400 | CA3 of left hippocampus | M1 | Left | Anterior hippocampus | Dual state | ±1.5 s | 20–50 Hz | 0.75 | 0.9 | 5ci |
| | Opto | 40 Hz sine wave | 8.09 | | 3 | 6 | 50 | 400 | CA3 of left hippocampus | M1 | Left | Hippocampus* | Dual state | ±2 s | 20–55 Hz | 0.75 | 0.6 | 5cii |
| | Opto | 40 Hz sine wave | 8.09 | | 3 | 6 | 50 | 400 | CA3 of left hippocampus | M1 | Left | Deep mesencephalic n. | Dual state | ±2 s | 20–55 Hz | 0.75 | 0.7 | 5ciii |
| | Opto | 40 Hz sine wave | 8.09 | | 3 | 6 | 50 | 400 | CA3 of left hippocampus | M1 | Left | Pulvinar n. of thalamus | Dual state | ±1.5 s | 20–50 Hz | 0.75 | 0.4 | 5civ |
| Postmortem stimulation | No stim | Empty room | | | | | | | | | | Power spectral density overlay of all channels from DC–600 Hz | | | | | | S3a |
| | No stim | No stimulation | | | | | | 400 | | M1 | | Power spectral density overlay of all channels from DC–600 Hz | | | | | | S3b |
| | Opto | 300 ms square waves | 9.53 | 300 | | 1 | 30 | 400 | CA3 of left hippocampus | M1 | | Power spectral density overlay of all channels from DC–600 Hz | | | | | | S3c |
| | Opto | 300 ms square waves | 9.53 | 300 | | 1 | 30 | 400 | Right arcuate cortex | M1 | | Power spectral density overlay of all channels from DC–600 Hz | | | | | | S3d |

**Functional mapping**. In addition to eliciting biomagnetic activity at the transduction site, optogenetic stimulation also elicited time-invariant activity in well-described downstream networks. One notable example of network activation was in response to right arcuate stimulation which elicited additional SAM peaks in a pattern that conformed to a motor-associative network, the cortico-striatal-pallido-thalamo-cortical (CSPTC) motor network[25,26]. This network was revealed in both NHPs, M1 and M2, who received arcuate stimulation across different optical stimuli (Fig. 4a; Table 1). An example of the CSPTC maps obtained for M1 is shown in Figure 4a. Briefly, in addition to the activation of right arcuate sulcus (ii), SAM maps also exhibited peaks in right anterior bank of the arcuate sulcus, right corona radiata/dorsolateral tip of the caudate n., right caudate n., right putamen/external segment of globus pallidus, posterior bank of contralateral (left) arcuate sulcus, right globus pallidus, right ventroposterior medial/ventroposterior lateral n. of thalamus, right medial dorsal n. of thalamus, ventral posterior thalamus, and parietal-occipital association area of the intraparietal sulcus (See Table 1 (CSPTC) for analysis details). Figure 4b represents a second example of right hemispheric CSPTC network activity for a different NHP, M2, in response to 20 Hz square wave pulse trains. Owing to the contrast of this subject's MRI, the thalamic peak identities were less discernible. Despite this, SAM maps likewise revealed extensive peaks throughout the CSPTC network in this second subject, M2 (See Table 1 for peak location and analysis parameters).

Similarly, left hippocampal stimulation also elicited activity throughout the temporal network. Figure 5a illustrates functional hippocampal and temporal networks engaged by a 20 Hz optical sine input to left hippocampus for M3. Whole-brain SAM SPMs for this stimulus exhibited peaks in ipsi and contralateral hippocampus as well as the ipsilateral amygdala, temporal cortex, and insula. (See Table 1 for peak locations and analysis parameters). A second example of hippocampal network activity in NHP M1 in response to 8 Hz sine wave input is presented in Fig. 5b. The SAM peaks elicited by this stimulus similarly respect known anatomical and functional relationships (See Table 1 for peak locations and analysis parameters). Figure 5c illustrates a third example of hippocampal network activity in response to 40 Hz sine wave input for M1. The SAM peaks are again consistent with known anatomical relationships (See Table 1 for peak location and analysis parameters). Collectively, these results indicate the utility of combining optogenetic techniques with MEG for functional brain mapping and mapping of deep structures, even under anesthesia.

## Discussion

We have developed a paradigm for whole-brain functional mapping based on high-resolution magnetic source imaging of optogenetically induced neural activity in the brains of vervet monkeys. In this report, we demonstrate the feasibility of functional brain mapping using this paradigm, and in doing so we provide empirical support for the use of MEG in the detection and analysis of signals with a very fine spatial resolution ($<1$ mm$^3$). This study represents the first ever combination of optogenetics with magnetic source imaging and provides additional "ground truth" information on the relationship between MEG signals and the underlying electrical activity in the brain[27,28].

Prior efforts at combining optogenetics with neuroimaging demonstrate the challenge of detecting the fast, highly localized, and transient activity associated with optogenetic stimuli. Using fMRI, Ohayon et al. used optogenetic methods to stimulate the frontal eye field in macaques, which did not elicit a detectable BOLD signal[29] until combined with simultaneous electrical stimulation of the same target (see also Dai et al.[30]).

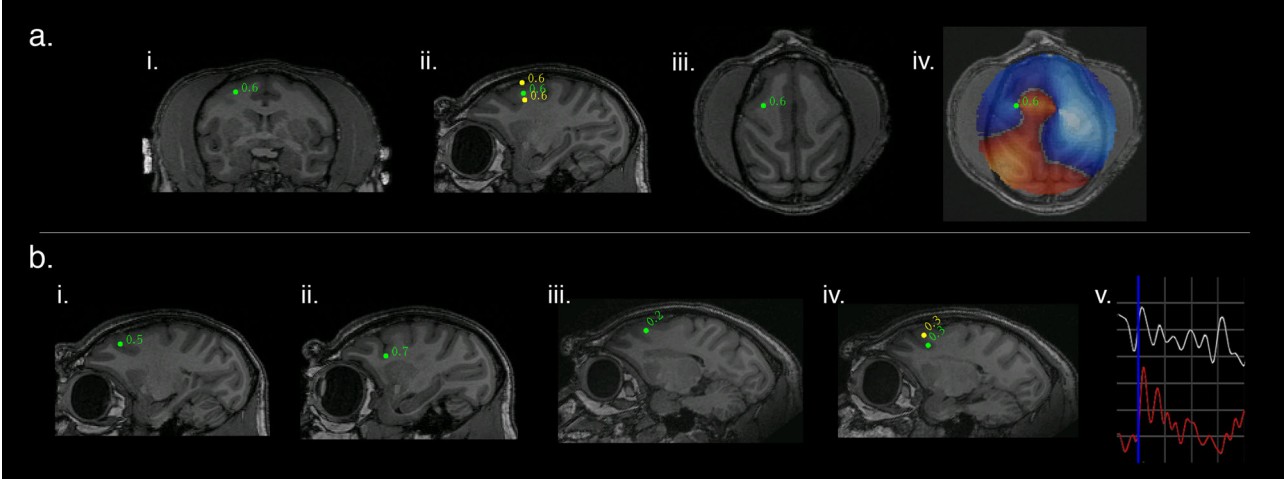

**Fig. 2 SAM source localization of optogenetically evoked MEG responses in the right arcuate sulcus in two representative animals and stimulation types. a** An example of optogenetic stimulation localized by dual-state SAM to the posterior bank of right arcuate sulcus for NHP M1 in the coronal (**i**), sagittal (**ii**), and axial planes (**iii**), for a stimulus of 50 ms square light pulses. The whole brain, un-thresholded, dual-state SAM SPM (voxel size of 750 $\mu m^3$, **iv**) reveals synchronized activity (red voxels) arising from the stimulation site and surrounded by desynchronization (blue voxels). Green dot and number indicate a peak in the SPM plus the associated pseudo-t value. **b** Right SAM arcuate peaks for four additional and different optical stimuli in two different NHPs. (**bi–ii**) present arcuate activations for M1 and (**iii-iv**) shows arcuate stimulation for M2. (**bi–ii**) is the same subject (M1) as presented in (**ai–iv**); (**i**) depicts the arcuate peak for 8 Hz sine waves, and (**ii**) shows the arcuate peak for 40 Hz sine waves. Arcuate stimulations in M2 are presented in (**iii–iv**); (**iii**) shows the arcuate peak for 10 ms square pulses and (**iv**) shows the arcuate peak for 20 Hz square wave pulse trains. A single trial (**v**) of an LFP recorded by the optrode and a simultaneous SAM virtual electrode for an arcuate peak (as seen in **biii**). Blue vertical line indicates stimulus onset (10 ms pulses). Both the LFP (white) and virtual electrode (red) exhibit a rapid peak following stimulation and have similar features and time courses. One gray square in the graph = 100 ms on the abscissa and for the ordinate, 35 $\mu$V for the LFP or 15 nA m for the virtual electrode. All maps follow radiological conventions.

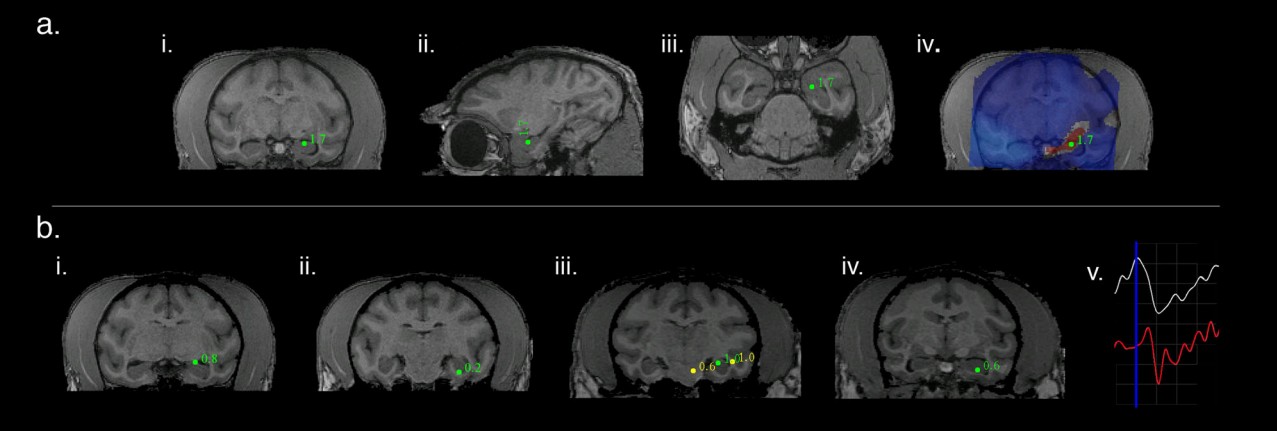

**Fig. 3 SAM source localization of optogenetically evoked MEG responses in the left hippocampus. a** SAM left hippocampal peak in coronal (**i**), sagittal (**ii**), and axial planes (**iii**) for a 20 Hz sine wave input in NHP M3. The whole brain, un-thresholded, SAM SPM shows synchronized activity arising from the stimulation site and surrounded by desynchronization (**iv**). **b** Left SAM hippocampal peaks for four additional and different stimuli in two different NHPs (M1 and M3). (**bi–ii**) is the same subject (M3) as presented in (**a**); (**i**) and (**ii**) show hippocampal peaks for single 150 ms ramp trials or 10 ms square pulses, respectively. (**biii–iv**) show hippocampal peaks for 8 Hz or 40 Hz sine waves, respectively, in the same subject (M1) as in Fig. 2a. A single trial (**v**) of a simultaneous LFP and SAM virtual electrode (both from 1 to 35 Hz) for hippocampus (as seen in **bi**). Blue vertical line indicates stimulus onset (ramp inputs). Both the LFP (white) and virtual electrode (red) peak rapidly following stimulation and have similar features and time courses. One gray square = 50 ms (abscissa) and 5 V for the LFP (amplified) or 500 nA m for the virtual electrode (ordinate).

In contrast, the temporal resolution of MEG enables the detection and mapping of fast, transient activity and interactions that have not yet been detected using other functional imaging approaches[31]. For example, here we stimulated arcuate cortex with trials of 50 ms square wave inputs, and beamforming revealed arcuate peaks at 75 ms with a voxel size of 750 $\mu m^3$ (Fig. 2a). Recent analysis techniques designed to increase fMRI sampling rate (e.g., dynamic functional magnetic resonance inverse imaging[32]), have achieved a maximum sampling rate around 10 Hz with a 5 mm³ voxel size[33,34]. The 10 Hz period corresponds to events lasting 100 ms, which remains well above the time scales relevant to the optogenetic response. Furthermore, we demonstrate that the recruitment of downstream areas can happen rapidly, suggesting that functional maps and connectivity analyses will vary depending on scan duration and sampling rate. In support of this, fMRI network architecture has been demonstrated to change over relatively long timescales (hours/minutes) during learning[35], while analysis of MEG based network

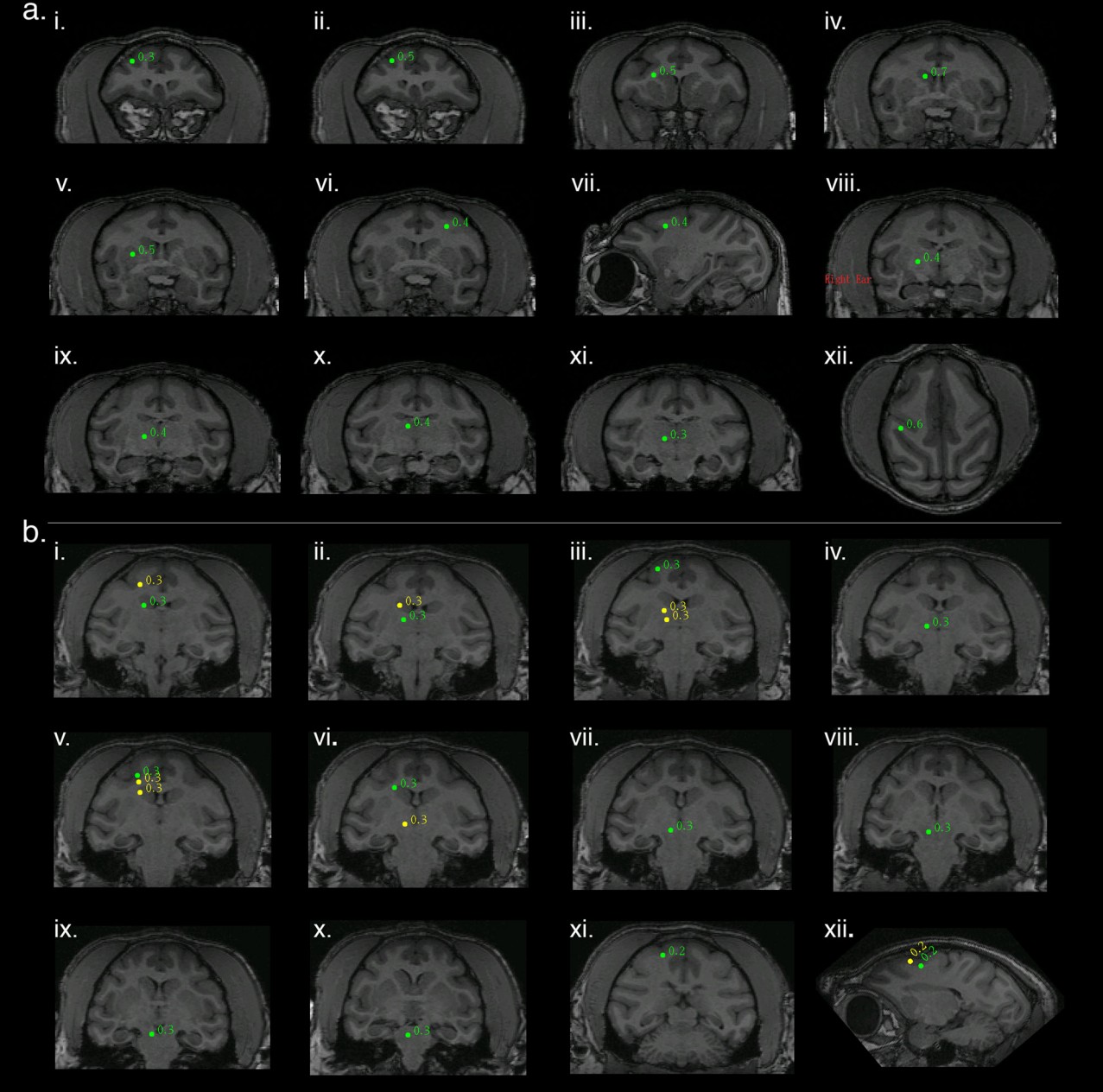

**Fig. 4 Arcuate optogenetic stimulation results in propagation through conserved networks across animals. a** Identification of cortico-striato-pallido-thalamo-cortical (CSPTC) network activity in response to 8 Hz sine wave stimulation of right arcuate sulcus in NHP M1. SAM analysis of optical input to arcuate sulcus reveals a known functional network with peaks in: (**i**) right inferior bank arcuate sulcus, (**ii**) right superior bank arcuate sulcus, the site of stimulation, (**iii**) right corona radiata/dorsolateral caudate n., (**iv**) right caudate n. (**v**) right putamen/ external segment of globus pallidus, (**vi**) and (**vii**) posterior bank of contralateral (left) arcuate sulcus, coronal and sagittal slices, respectively, (**viii**) right globus pallidus, (**ix**) right ventroposterior medial/ ventroposterior lateral n. of thalamus, (**x**) right medial dorsal n. of thalamus, (**xi**) right ventral posterior thalamus, and (**xii**) parietal-occipital association area of the intraparietal sulcus. **b** A second example of right hemispheric CSPTC network activity in a different NHP (M2) in response to 20 Hz square wave pulse trains with SAM peaks as follows: **i** caudate n. (green) and cortical area 3 (yellow), (**ii**) thalamus (green) and caudate n. (yellow), (**iii**) posterior dorsal bank of the arcuate sulcus (green) and two thalamic peaks (yellow), (**iv**) thalamus, (**v**) posterior ventral bank of arcuate sulcus (green, site of stimulation), corpus callosum (yellow), and caudate n. (yellow), (**vi**) caudate n. (green) and mesencephalic nuclei (yellow), (**vii**) mesencephalic n., (**viii**) substantia nigra, (**ix**) tegmental n., (**x**) pontine or mesencephalic n., and (**xi**), and (**xii**) all peaks in primary motor cortex (area 4).

architecture has demonstrated variability on the order of milliseconds[36]. It is clear from our demonstration that the high sampling rates of MEG coupled with SAM affords particular advantages in spatiotemporal resolution over traditional neuroimaging techniques. The combination of MEG and optogenetics provides a useful platform for the exploration of brain dynamics

across a broad range of the relevant temporal and spatial scales at which the brain operates.

Our study is a significant extension and enhancement of MEG for nonhuman primate research. Nonhuman primates provide versatile and robust models of brain function for both healthy and diseased states—with excellent face validity for recapitulating

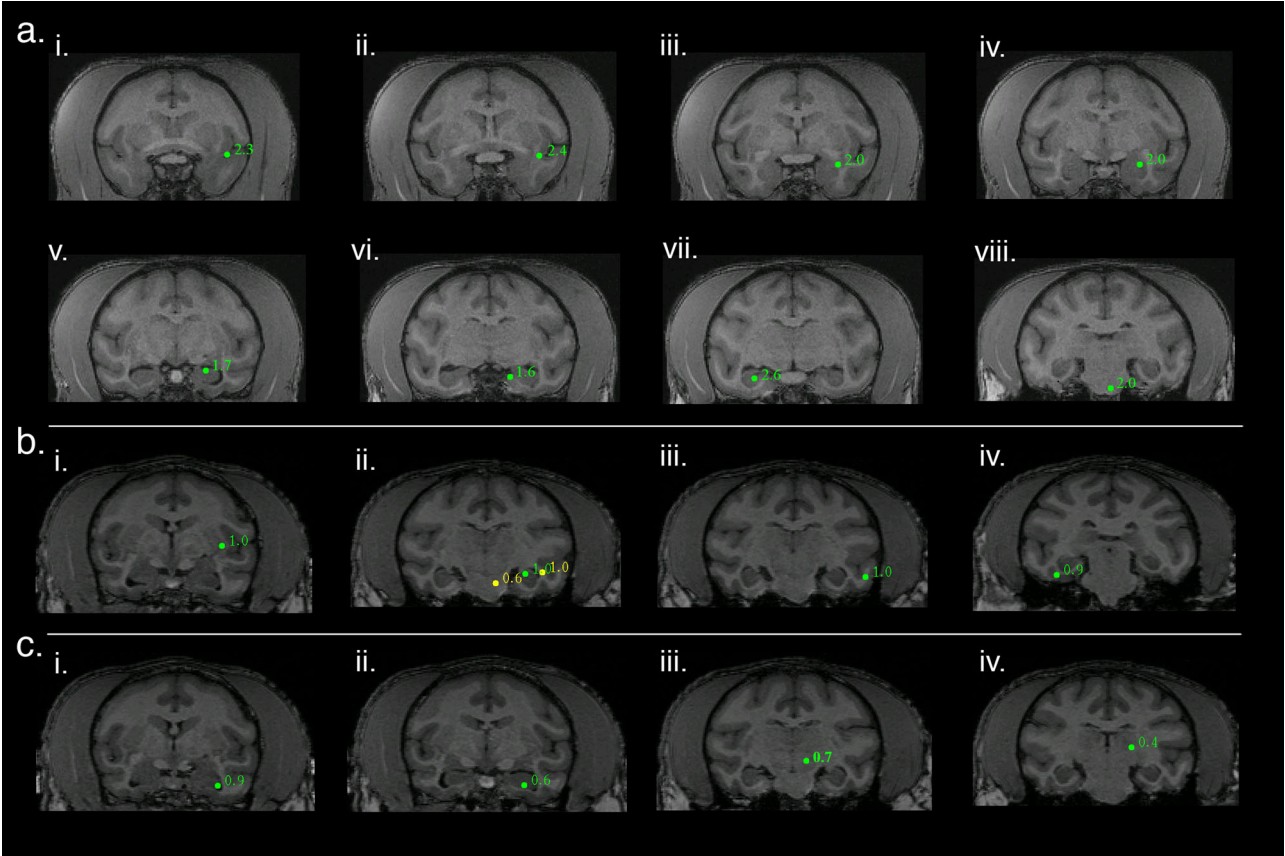

**Fig. 5 Hippocampal optogenetic stimulation results in propagation through downstream networks across animals. a** Functional hippocampal and temporal networks in NHP M3 engaged by a 20 Hz optical sine input to left hippocampus (**v**) with SAM peaks as follows: (**i**) left insular proisocortex/ temporopolar proisocortex, (**ii**) left para-insular cortex, (**iii**) left dorsolateral amygdala (lateral n.), (**iv**) left basolateral/lateral amygdala, (**v**) left hippocampus, site of stimulation, (**vi**) left parasubiculum/presubiculum of hippocampus, (**vii**) contralateral (right) hippocampus, and (**viii**) left pontine reticular activating formation. **b** A second example of hippocampal network activity in a different NHP (M1) in response to 8 Hz sine wave input with SAM peaks as follows: (**i**) left putamen, (**ii**) PGa/IPa association areas of left temporal cortex (lateral yellow peak), left hippocampus (green peak, site of stimulation), and possible left red n. (yellow, medial peak), (**iii**) area TEA/TEM of left temporal cortex, and (**iv**) area TL/TFM of right temporal cortex. **c** A third example of hippocampal network activity in response to 40 Hz sine wave input (same subject as in **b**) with SAM peaks as follows: (**i**) left anterior hippocampus, (**ii**) left hippocampus, site of stimulation, (**iii**) left deep mesencephalic n., and (**iv**) left pulvinar n. of thalamus.

human pathologies. Despite the advantages of NHP models in studying brain function, perceived hurdles to the utilization of NHP MEG preparations include the concern that the NHP head is small and therefore too distant from most of the MEG sensors to generate detectable biomagnetic signals. Accordingly, only four MEG studies have been carried out in NHPs[37–40].

This assumed barrier is attributed to the decay of magnetic fields at a rate inversely proportional to the cube of distance[2]. A second and related problem is that there are conflicting reports as to whether MEG can resolve activity from deep structures in general, and Mikuni et al. estimated that 4 cm² of active brain tissue is required for the localization of brief events[3], further limiting the assumed applicability of MEG studies. Our results show that very focal, fast-transient signals are detectable throughout the brain volume, and that significantly less tissue is required to generate a detectable MEG signal than was previously suggested.

In this study, SAM detection and localization were possible even in the presence of high noise[13,22] and suboptimal conditions resulting from the experimental preparation (e.g., the small size of the vervet brain, a large heartbeat artifact, artifact from anesthesia monitoring equipment, etc.). While the amplitude of the localized signal (as given by SAM pseudo-t score) varies, this is likely due to the type of stimulus input and can be useful in the future to

examine the effects of different time-varying inputs into a target structure. Additionally, because inverse methods are without unique solutions there will always be inherent uncertainties in localization[41] without "ground truth" experiments utilizing controlled inputs to known brain targets. To this end, the approach we report may assist in efforts to continually to improve inverse solutions to yield greater certainty[42,43].

We show that after optogenetic stimulation, both arcuate cortex and hippocampus can be localized by SAM across multiple NHPs, across stimulus types, and at a voxel size of only 750 μm³. Studies using nonhuman primates in MEG have previously demonstrated the feasibility of equivalent current dipole[37] and beamformer[38] analyses in monkeys. This study extends these observations to analysis of areas beyond superficial cortex into deep structures in an NHP species with a smaller brain, strongly supporting the validity of SAM in detecting deep brain signals with high resolution. In fact, consistent with SAM theory[2,6,8,10], we demonstrate that the ability to detect focal, fast-transient signals is actually degraded by larger voxel sizes (Supplementary Fig. 2).

We have demonstrated that optogenetics can be effectively combined with MEG imaging in non-human primates to permit precise functional mapping. As we report, for example, stimulation of arcuate sulcus activated the cortico-striato-pallido-

thalamo-cortical network such that voxels of peak activation fell within a functionally and anatomically described pathway[25,26]. Additionally, hippocampal stimulation evoked widespread activity in regions with known connections to hippocampus such as amygdala, insula, and contralateral hippocampus. Finally, natural visual stimulation using flashes of light elicited activity throughout the visual system. Moving forward, the combination of optogenetic techniques with MEG and beamforming could be used to variably activate or inactivate discrete parts of the brain with excitatory or inhibitory opsins to reveal functional relationships that have not previously been described. We also anticipate that experimenter control over discrete brain regions of interest coupled to whole-brain recording and detection of downstream responders will provide a platform for improved investigation of brain function in both normal and pathological conditions in NHP models.

## Methods

**Surgical targeting**. All studies complied with ethical standards for the use of research animals and were reviewed and approved by the Animal Care and Use Committee of Wake Forest School of Medicine. This study was repeated in three female vervet monkeys, *Chlorocebus aethiops*, aged 15-16 years and weighing between 4.99 and 6.75 kg. Each animal was mounted in an MRI compatible stereotaxic frame with lipid containing ear bars, and T1-weighted anatomical images were acquired using a 3D-MPRAGE 0.5 mm isotropic sequence in a Siemens 3 T Skyra system with a custom built 8-channel flexible head coil (Dr. Cecil Hayes, Univ. Washington). The animals were removed from the stereotaxic frame and an additional structural MRI scan was acquired in which the animals were fitted with three lipid biomarkers fixed in the location of the MEG fiducials (placed in the conventional three-point fiducial locations) for co-registration with MEG data[44]. The locations of the MEG fiducials were tattooed on the animals prior to the start of the study to ensure precise co-registration across multiple sessions. In one animal a postmortem MRI was conducted after implantation and recording to confirm stereotaxic targeting (Fig. 1aii–iii and bii–iii).

Using Medical Image Processing, Analysis, and Visualization (MIPAV) software available from the NIH, stereotaxic coordinates were generated for CA3 of left hippocampus and for the posterior wall of the contralateral (right) arcuate sulcus.

**AAV vectors**. AAV2/10-CaMKIIa-ChR2-eYFP was provided as a gift by Dr. Caroline Bass. The AAV vector titer was between $1 \times 10^{12}–1 \times 10^{13}$ GC/ml. The AAV vector used in this study was created using AAV helper plasmid which allows for high purity, high titer vectors[45].

**Surgery**. Under isoflurane anesthesia, animals were securely mounted and centered in a surgical stereotaxic frame. The cranium was exposed and ten ceramic screws (Rogue Research Inc.; Montreal, Canada) were placed around the perimeter of the surgical margin. A craniotomy (~17 mm) was then drilled over the coordinates for each targeted brain region. A custom polyether ether ketone (PEEK) cannula with a Teflon stop was advanced through the dura on a stainless steel stylette. The stylette was retracted leaving the cannula in place; injection of the viral vector and placement of the optrode (optical fiber and recording electrode) was done through the cannula to ensure co-localization. A gas-tight, micro volume Hamilton Syringe was then filled with 4 μL of virus which was delivered in a single injection at each target area through a 38 G needle at a rate of 0.5 μL/minute. The syringe was left in place for 5 min following the injection and slowly removed. A custom made, chronically indwelling optrode was implanted to a depth of 0.5 mm above the injection coordinates. The optrode consisted of either a 200 or 400 μm diameter fiber optic cable (ThorLabs, Inc.; Newton, NJ) coupled to a 75 μm Pt/Ir electrode (FHC, Inc.; Bowdoin, ME) or a 35 μm formvar-coated Tungsten electrode (California Fine Wire Co.; Grover Beach, CA), which extended 0.7 mm beyond the tip of the optical fiber[46]. Lastly, a custom (PEEK) headwell with a removable cap (Crist Instruments Co., Inc.; Bethesda, MD) was fitted over each craniotomy and cemented in place. The craniotomy was sealed with bone cement securing the optrode in place.

**Photovoltaic effect**. All optogenetic studies that utilize concurrent electrophysiological recordings must take steps to avoid the photovoltaic effect whereby intense light shining directly on an exposed metal surface submerged in saline can produce a measurable voltage[47]. Accordingly, we avoided the photovoltaic effect by designing optrodes with a beveled tip so that the conductive surface was in shadow rather than in the cone of light. The effectiveness of this strategy was confirmed in saline and in pre-expression control recordings (Fig. S4). Prior to implantation, a combined fiberoptic/recording electrode (optrode) was submerged in saline and LED stimulation was conducted to detect artifacts associated with the photovoltaic

effect. No transient artifact associated with the LED was detected upon stimulation and there was no significant difference in LFP amplitude from the pre-stimulation period suggesting the optrode design used was successful at reducing the chance of photovoltaic artifacts.

At the conclusion of the study, one animal was euthanized to confirm electrode placement (see Histology section). Prior to the histological examination, the subject was scanned postmortem as an MEG/MSI negative control to rule out the influence of photovoltaic field effects on the biomagnetic signal. The hippocampus and arcuate cortex were each stimulated separately with 300 ms square pulses of light for 150 trials and with a 1 s interstimulus interval. The power spectra of the MEG channels during the postmortem stimulation were not different than the power spectra obtained from an empty room (Fig. S3). Dual-state SAM beamforming was conducted for ±300 ms windows relative to stimulus onset, with a high-pass filter set at 4 Hz to prevent aliasing, and the low-pass filters were varied in a series of 10 Hz steps ranging from 50 to 90 Hz (see Analysis for general beamforming details). SAM did not reveal any peaks in either arcuate cortex or hippocampus at any of the frequency bands imaged, suggesting that the photovoltaic effect did not contribute to the biomagnetic signal during stimulation trials.

**Electrophysiology recordings**. Recordings were conducted under anesthesia, using propofol (3.0–4.0 mg/kg induction; 200–400 μg/kg/min via syringe pump [Sage, Orion Research Corporation, Cambridge, Mass]) for the MEG scans and ketamine (12–15 mg/kg) and dexmedetomidine (0.0075–0.015 mg/kg) for the electrophysiological recordings. The animals were intubated and artificially ventilated, which permitted supine positioning of the monkeys in the MEG scanner and which minimized head movements. MEG recordings were conducted in a magnetically shielded room (MSR; Vacuumschmelze GmbH & Co.; Hanau, Germany) with a CTF MEG™ whole cortex helmet equipped with 275 first-order axial gradiometer coils, each with a 5 cm baseline and 22.4 mm average inter-sensor spacing and 29 reference sensors (CTF MEG International Services Limited Partnership; Coquitlam, BC, Canada). MEG recordings were sampled at 2400 Hz for a bandwidth of DC-600 Hz. Data were powerline filtered offline for 60 Hz harmonics with a 4 Hz width, and synthetic third-order gradiometry was applied. Simultaneously recorded LFP data were amplified using a Nicolet intraoperative monitoring system (Natus Medical; Pleasanton, CA) sampled at 5 kHz with a maximum range of ±5 mV with a bandpass filter of 0.1 Hz to 3 kHz and recorded in line with MEG data. On separate days without MEG recordings, LFP data were recorded using a SciWorks recording system (DataWave Technologies; Loveland, CO), AM-3600 extracellular amplifiers (A-M Systems; Carlsborg, WA), and a T8G100 headstage amplifier (Triad Biosystems International; Durham, NC) and bandpass filtered at 0.3 Hz to 5 kHz, with a gain of 500 and a 40 kHz sampling rate[48].

Prior to the optrode implantation, 8-minute resting-state MEG scans were acquired to establish baseline activity. Two weeks after surgery (before high-level transgene expression, and after the animal had recovered from surgery) animals were stimulated for the first time at a range of optical intensities (10.3–75.6 mW/mm² for 200 μm and 2.6–18.9 mW/mm² for 400 μm fibers). The spatial extent of light spread from the fiber tip was estimated to be 0.45–1.65 mm (>0.1 mW/mm²) for the 200 μm fiber and 0.3–1.9 mm for the 400 μm fiber, using an empirically derived model[49]. These recordings were conducted in the MEG MSR under anesthesia as described, as well as in separate sessions in a dedicated electrophysiology suite. A variety of optical stimuli were delivered, including single 10 ms or 50 ms square pulses of 473 nm light from either a light-emitting diode (ThorLabs) or from a laser (Shanghai Dream Laser Technology Co., Ltd; Song Jiang, Shanghai, China), 20 Hz square wave pulse trains with a 4 ms pulse width, single ramps, and sine waves at 8, 20, or 40 Hz (see below for more stimulus details). The pulse duration of the laser was controlled with a 2 mm Uniblitz laser shutter and a D880C Uni-stable driver (Uniblitz Electronic; Rochester, NY). LED pulse durations were controlled with custom MATLAB scripts (Natick, Massachusetts: The MathWorks Inc.) and a digital–analog converter (DAC; Data Translation, Inc.; Marlborough, MA). Prolonged high power optical stimulation can produce heating and change the firing properties of neurons even in the absence of optogenetic proteins[50]. All of the stimuli selected for investigation here were optimized to avoid heating tissue by using either low duty cycle (short pulses) or short-duration trains (3 s), with ample interstimulus intervals (6 s). After stable transgene expression was determined, the animals were stimulated again using the same paradigms as above and the resulting activity was recorded. Electrophysiological and MEG recordings after stable expression were again done simultaneously using the Nicolet system for acquiring electrophysiology during MEG and also separately.

We conducted an analysis of visual stimulation as a positive control for beamformer accuracy by presenting pulses of white light via fiber optic cable to the left lower quadrant of the visual field of the left eye of the sedated animal. The right eye was taped shut per typical recording protocol while the left eye remained open. The fiber optic cable was channeled into the helmet slightly lateral and inferior to the left eye.

**Analysis**. All analyzed data were derived from recordings in which motion was no greater than 0.2 mm. Head motion was nearly eliminated with anesthesia and a carefully supported head.

We performed data preprocessing, head model creation, and beamforming using CTF MEG™ Software (CTF MEG International Services Limited Partnership, Coquitlam, BC, Canada). MEG preprocessing included DC-offsetting, application of synthetic third-order gradiometry, and powerline filtering[13,22]. MEG data were then co-registered with the monkey's anatomical MRI data using the standard three-point fiducials. From this, a multiple-overlapping-spheres model of the head and whole-brain volume was generated[51]. Dual-state SAM[8,10] was applied to each active time segment and an equal length control segment composed of data preceding the stimulus, to construct 3D images of the entire brain volume. In this case, biomagnetic signals during active-state trials are compared to signals in pre-stimulus, or baseline trials and are expressed as a ratio of power in a given frequency band and on a per-voxel basis. This ratio also inherently contains estimates of background/environmental/common mode noise in both active and baseline conditions, and this noise (in addition to trial-to-trial signal fluctuations) can also be taken as a measure of variance. Thus, given the degrees of freedom (number of trials, number of voxels) the pseudo-t metric is computationally very similar to the t-statistic but more efficient to calculate[2,8], and has been utilized to map task-related activity[2,52]. Noise normalized pseudo-t—statistical parametric maps of source power were derived from the beamformer output at a voxel size ranging from 0.75 to 2.0 mm³. Results are shown for 750 μm³ analyses due to superior accuracy in source localization. Such small voxel sizes were possible because of the extremely low motion associated with this preparation. The degree of activation in each voxel is indicated by a pseudo-t-score. Local maxima (synchronization, red) and minima (desynchronization, blue) of the SAM maps were identified as voxels of peak activity.

The effects on localization by changing voxel size are presented in Fig. S2 (analysis details in Table 1 -Voxel Size), and the maps are an extension of the data presented in Fig. 2a. At the finest level of analysis (750 μm³) three sources (peak voxels/maxima) were separable[8] along a narrow band of event-related desynchronization (blue) in the map. As the voxel size was increased to 1 mm³, two of the sources blurred to form a single peak with pseudo-t = 0.6, while the most ventral source dropped in power. For a voxel size of 1.5 mm³ (S2c), the three arcuate peaks formerly seen in S2a have merged into a single peak with pseudo-t = 0.6, and for a voxel size of 2 mm³ the SAM map is unfocused, and no arcuate sources can be resolved. Similarly, the narrow band of desynchronization as seen in S2a broadens as the voxel size increases, such that at the limit of 2 mm³ no sources can be identified. These results indicate the utility of beamforming with very small voxel sizes given computational and motion constraints.

SAM time-frequency analysis parameters were determined in part by the time-frequency characteristics of the input stimulus and by the power spectral densities (PSDs) and timing of the neural response measured on the LFP during simultaneous MEG/LFP recordings in the MSR when such recordings were obtained. In general, for brief stimuli, the beamforming windows necessary to detect activity changes also needed to be short, with a correspondingly wider frequency band to provide enough samples for analysis. For the sine wave inputs, the SAM frequency bands were set to bracket the sine frequency, and because the sine inputs were of longer duration the beamforming windows were correspondingly longer, both to provide enough samples for the analysis and also because the brain circuits appeared to sustain activity for a long time (see Results). Specific arcuate beamforming parameters for each stimulus type are as follows. For the 50 ms square pulses (120 trials, 5 s ISI), dual-state SAM relative to stimulus onset was conducted in 50 ms steps from 100 to 500 ms inclusive and for another window of 75 ms (Fig. 2ai–iv). For each time window, the high-pass was set above the threshold for aliasing and the low-passes were varied in a series of 10 Hz steps ranging from 40 to 100 Hz and another step at 35 Hz. Peak voxels were identified in each whole-brain, time-frequency map and because peak voxels often appeared in multiple maps the time-frequency combination that expressed the maximal pseudo-t score for each peak was chosen as the representative map. A primate neuroanatomist (J.D.) confirmed the neuroanatomic localizations of the peaks to their target structures and also for downstream structures that were also activated following stimulation. For the 8 Hz sine waves (3 s stimulus length, 20 trials, 10 s ISI, Fig. 2bi) the SAM time-frequency parameters were 1–4 s windows in 500 ms steps, a DC high-pass, and low-pass steps of 12, 15, 20, and 40 Hz. For the 40 Hz sine waves (3 s stimulus length, 20 trials, 10 s ISI, Fig. 2bii) the SAM parameters were 500 ms–4 s windows in 500 ms steps, a high-pass series ranging from 10 to 35 Hz in 5 Hz steps, and low-pass steps at 50 and 55 Hz. For the 10 ms square pulses (100 trials, 6 s ISI, Fig. 2biii), SAM was conducted in 50 ms steps for windows from 100 to 750 ms, plus additional windows at 125 ms and 1 s, while the frequency bands had a minimum high-pass set to prevent aliasing for each time window, and a low-pass set at 50, 70, and 100 Hz. For the 20 Hz square pulse train (5 ms width squares, 10 pulses/train, 100 trains, 6 s ISI, Fig. 2biv) the SAM time-frequency parameters included windows from 100 to 500 ms conducted in 100 ms steps, plus an extra window at 750 ms, a high-pass set to prevent aliasing for each window, and a low-pass at either 70 or 100 Hz.

Beamforming parameters for the hippocampal stimulation experiments are as follows. For the single 60 s, 20 Hz pulse train (9.53 mW, Fig. 3ai–iv) the dual-state SAM time windows were 2, 3, 4, 5, 10, 20, 30, and 60 s, the high-pass was either 10 or 15 Hz, and the low-pass was 30 Hz. For the single ramps (150 ms duration, 20 trials at an intensity of 9.53 mW, 6 s ISI, Fig. 3bi) the SAM parameters included 150–400 ms time windows in 50 ms steps, plus additional windows at 125 ms and 500 ms, a high-pass set to prevent aliasing per time window, and low-passes ranging from 40 to 55 Hz in 5 Hz steps, plus additional windows at 70, 100, and 120 Hz, plus at 35 Hz for windows of sufficiently long duration. For the 10 ms square pulse stimuli (114 trials, 9.2 mW, 6 s ISI, Fig. 3bii) the SAM time-frequency parameters included 50, 175 ms windows, in 25 ms steps, a high-pass set to prevent aliasing per time window, and low-passes ranging from 30 to 100 Hz (when the time window permitted a small low pass) in 5 Hz steps, with additional windows at 120 and 200 Hz. For the 8 Hz sine waves (3 s duration, 20 trials, 10 s ISI, Fig. 3biii) the beamforming parameters included 1–3 s windows in 500 ms steps, a DC high-pass, and a low-pass of 12, 15, or 20 Hz. For the 40 Hz sine waves (3 s duration, 20 trials, 10 s ISI, Fig. 3biv) the beamforming parameters included 1–3 s windows in 500 ms steps, high-passes of 10, 20, 30 or 35 Hz, and low-passes of 50, 55, or 60 Hz.

Beamforming for the visual stimulation experiment (Fig. S1) included 50–300 ms windows in 25 ms steps, a high-pass set to prevent aliasing per time window, and low-passes ranging from 30 to 70 Hz in 5 Hz steps, plus additional windows at 80, 90, 100, 120, and 200 Hz.

LFP data were analyzed using SciWorks, Neuroexplorer (Nex Technologies, Madison, Al), and custom scripts in MATLAB. The peak amplitude of the LFP in response to optical stimulation during electrophysiology recordings was measured between two-week, five-week, and seven-week time points. Peak amplitudes for each replicate were measured at the time point corresponding to the peak of the average trace for each light intensity.

**Histology**. To confirm the anatomical position of the optrode placement and injection site, at the completion of the imaging study, one animal was necropsied by first sedating with ketamine followed by an overdose with pentobarbital. A thoracotomy was conducted and the animal was transcardially perfused with ice-cold phosphate-buffered saline followed by 4% paraformaldehyde fixative. The brain was extracted and sectioned at a 50 μm thickness on a modified Vibratome 1000 (Leica Biosystems Inc.; Buffalo Grove, IL). Sections of tissue were mounted on gelatin-coated microscope slides and imaged on a Zeiss LSM 710 confocal microscope with ×10 and ×20 objectives (Carl Zeiss Microscopy, LLC; Thornwood, NY).

**Reporting summary**. Further information on research design is available in the Nature Research Reporting Summary linked to this article.

## Data availability
The MEG, electrophysiology, and histology data generated in this study are under active use by the reporting laboratory; all data presented in this manuscript are available by reasonable request. The CTF-formatted MRI data and CTF MEG SAM maps from which Figs. 2–5 were generated are available at the following link: https://doi.org/10.5281/zenodo.5148994. Source data are provided with this paper.

## Code availability
Custom MatLab scripts are available by request and at: github.com/neuroptics/optoDR.

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

## Acknowledgements

C.C. was supported by NIH grant R01 MH097695. J.B.D. was supported by NIH grant NIAAA R21 AA028007, P01AA021099-S1, P50AA026117, UL1TR001420, and internal CTSI Ignition funds. D.W.G. was supported by NIH Grant NIAAA R01AA016852, P50AA026117, W81XWH-13-2-0095, NINDS 1R21NS116519, Neuroscience Clinical Trials and Innovation Center, the WFU Clinical and Translational Science Institute, and UL1TR001420. D.K. was supported by NIH grants NIAAA R01AA016852, NINDS 1RO1NS105005, and NINDS 1R21NS116519. G.E.A. was supported by NIAAA F30 AA 23708-02 and NINDS NS073553. J.S.K. was supported by VA/DoD W81XWH-13-2-0095, NIH grant NIAAA R21 AA028007, and internal CTSI Ignition funds. We wish to thank the Department of Neurology and Dr. Gautam Popli for providing scanner time on the MEG and the Translational Imaging Program (then Center for Biomolecular Imaging) for MRI pilot scans. We acknowledge the Wake Forest University Primate Center for providing animals for this study OD010965 (Jay Kaplan) and the Wake Forest Biology Microscopic Imaging Core Facility. We also acknowledge support from the W.G. (Bill) Hefner Veterans Affairs Medical Center and VA Mid-Atlantic Mental Illness, Research, Education, and Clinical Center. Finally, we would like to thank Dr. Caroline Bass for providing the viral construct used in this study.

## Author contributions

G.E.A. contributed to experimental design, data collection, data analysis, interpretation, and manuscript preparation. J.R.S.K. contributed to MEG data analysis, interpretation, and manuscript preparation. D.C.K. contributed to experimental design, data collection, electrophysiology data analysis, interpretation, and manuscript preparation. E.R.R. contributed to data collection and manuscript preparation. C.C. contributed to experimental design and manuscript preparation. J.B.D. contributed to experimental design, data collection, data analysis, and manuscript preparation. D.W.G. contributed to experimental design, data collection, interpretation, and manuscript preparation.

## Competing interests

The authors declare no competing interests.
