## [Peer Review File · Nature Communications]

Simultaneous optogenetic activation and MEG source imaging show non-human primate brain circuitsREVIEWER COMMENTS

Reviewer #1 (Remarks to the Author):

The authors aimed to develop an experimental platform combining MEG-compatible optogenetic techniques in non-human primates (NHPs) to test the ability of MEG/MSI to image deep signals. The study concludes that this is the first demonstration of accurate localization of deep sources within an intact brain using a novel combination of optogenetics with MEG/MSI. This is a very interesting study with clear methodological implications for future investigations. However, I have several major and minor concerns which I feel should be addressed via major revisions before this manuscript would be suitable for publication. One of the secondary goals is to provide evidence for the validity of using MEG to localize deep hippocampal sources. Please find my comments below:

Major:

1) Conceptual: There are conceptual aspects of the study that have not been clearly articulated for readers who are non-experts in the methods described in this manuscript, such as myself. I feel that the Introduction section does not provide a compelling rationale for conducting the study, nor what the study's implications might be going forward (other than those clinical in the field of epilepsy). For example, what is the significance of providing "a detailed investigation of the resolution capabilities" of MEG? What would be the implications of doing so?

2) The paper does not flow well from rationale, to unknown or hypotheses to results and specific conclusions that address the unknown or hypothesis. It comes across as a "tour de force" in the combination of various techniques but is unclear why were they use in this specific way in the first place. For example, it is stated as important that it is possible to use human MEG to evaluate function in this particular animal but why is that important?

3) Analysis: There is not an adequate rationale given for some aspects of the analyses and thus leads this reader to doubt the validity of the study conclusions. Please make clear why the pseudo t-score was used. Was it planned that pseudo t-scores would be used, and if not, why do the authors think that there was increased noise? This same question goes for use of unthresholded images. Along these lines, the authors should add a limitations section.

4) Presentation of results: The results should be presented in a more organized manner as opposed to long in-text descriptions. For example, the information described in Lines 178-212 could be supplied in a table. In addition, some of the results presented in-text and in figures are largely duplicable rather than complementary. One such instance includes portions of Lines 149-152 ("The blue vertical line indicates the time of stimulus onset, in this case a 10 ms square wave pulse. Both the LFP and virtual electrode exhibit a rapid rise and peak in activity following stimulus onset, and both qualitatively share similar features and time courses on a single trial basis."), which are almost exactly the same as the figure description in Lines 656-657 for Figure 2 ("Blue vertical line indicates stimulus onset (10 ms pulses). Both the LFP (white) and virtual electrode (red) exhibit a rapid peak following stimulation and have similar features and time courses.").

Minor:

1) Lines 79-81: What units do these CI's relate to?

2) Lines 250-251: add "to" after "can be used".

3) Line 254: But others have already used MEG to detect signals from deep brain structures, as noted in the Introduction section.

4) Line 361: Can the authors provide a reference for the conventional three-point fiducial locations?

In summary, material in this paper appears valuable as a technical contribution of possible interest for clinicians and basic science investigators. The relatively disorganized narrative reduces enthusiasm. The paper could gain from a more structured presentation format, focusing around the resolution of a primary and perhaps secondary questions. The main take-home messages appears diluted by various claims that leave the read somewhat confused.

Reviewer #2 (Remarks to the Author):

Alberto and colleagues used viral vector-mediated optogenetics in anesthetized vervet monkeys to

assess the ability of MEG-based techniques to detect changes in neural activity in the cortex and hippocampus. They found that optical stimulation of ChR2-expressing sites elicited MEG activity near the injection sites and in several downstream areas. Optical stimulation of transduced sites in the arcuate sulcus elicited activity in regions of the cortico-striatal-pallido-thalamocortical motor network, whereas optical stimulation of CA3 elicited activity in the temporal network. The authors conclude that this union of techniques (optogenetics and MEG/MSI) will be useful for exploring causal functional relationships between brain areas.

The manuscript is a valuable addition to the literature, although the novelty of the results should be articulated more clearly. Optogenetics has already been combined with fMRI in the macaque monkey (Ohayon et al., *Journal of Neuroscience*, 2013). Interestingly, in this previous study, optical stimulation of a cortical region (FEF) near the arcuate sulcus failed to elicit fMRI activity whereas electrical stimulation in combination with optical stimulation did. This relevant previous study is not cited or leveraged in the current manuscript. I believe that the authors are missing a unique opportunity to draw on similarities/differences between the two studies and between the resolution of MEG/fMRI to strengthen their claims of novelty.

Aside from the above concern, there are other concerns related to methodology and interpretation that should be addressed. These points are outlined below.

Major concerns

- Several important details about the AAV vector injections are missing or unclear. What was the titer of this virus? And is there a reason the authors chose the AAV10 capsid serotype? Did the authors inject only 4 microliters of vector at one site in each area? Did the authors deliver the vector at several depths in each area?
- The histological images in Figure 1 are hard to evaluate. Is the green signal shown native fluorescence or was immunohistochemistry performed? Can the authors provide higher magnification images that clearly show transduced cell bodies in each area? And how long after the vector injection was this animal terminated? Finally, did the authors attempt to relate the spatial extent of vector transduction in this animal assessed histologically to the spatial extent of the MEG signal elicited in this animal?
- The authors use a variety of optical stimulation protocols (square, sine, saw-tooth) and different laser modulation frequencies. However, the authors never explain the rationale for these different stimulation protocols or their utility for performing functional mapping studies with MEG. It is currently unclear to the reader if/why the different stimulation protocols or wavelengths make a difference.
- The supplementary figure shows a collection of brain regions engaged by visual stimulation through the eye. I'm confused about why this visual stimulus doesn't elicit activity in ALL visual areas (both cortical and subcortical). Is this a limitation of the MEG/MSI technique? Or is this an effect of anesthesia? Or are the authors only selecting to show a subset of the areas activated?
- It would be useful to give readers a sense of how the magnitude of the LFP and/or MEG signal induced by visual stimulation compares to that induced by optical stimulation.
- Did the authors vary the intensity of optical stimulation and attempt to estimate the ability to detect neural activity with MEG? It could be useful to estimate the lowest light intensity that elicits MEG activity and how that might relate to the spatial resolution of the MEG signal.
- The section on "Functional mapping" reads like an exhaustive list of downstream areas where MEG activity is detected following optical stimulation near the injection site. What should the reader take away from this list? Are there areas that were expected to be connected to the arcuate sulcus and CA3 that do not show up in the MEG analyses?

Minor concerns

- The authors use the words "transfected/transfection" incorrectly (Lines 72, 76, 93, 121, 178). "Transfection" typically refers to gene delivery via a plasmid not a virus. "Infection" refers to gene delivery via a virus, which is the case for this manuscript. However, it is more common in studies of viral vector-mediated optogenetics to use the words "transduced/transduction", which are equivalent in meaning to "infection". It would be best if the authors revisit their usage, perhaps referring to sites of interest as "transduced sites" or simply "injection sites"?
- Figure 1: Panel C: It would be more appropriate to label these data based on the number of weeks post-injection rather than "pre-expression" and "post-expression". We cannot be entirely sure that there is no expression at 2 weeks; we only know that there is no detectable activation at 2 weeks. Panels D and E: Are these data from a single session in one animal - or multiple sessions and animals?
- As a non-expert in MEG analyses, it is unclear to me what the "virtual electrode" analysis entails and whether this is a widely used measure in the field. Relatedly, it is unclear to me whether the way that the authors represent SAM/MEG results in Figures 2–5 (single points with weights) is common in the MEG literature. It might be useful to comment on these points to accommodate broader readership.
- Line 54: Not all opsins are light sensitive "ion channels"—some are pumps.
- Line 435: I think there is a typo here. Do the authors mean "in vivo" not "in vitro"? And how does one test the optrode in post mortem tissue? This sentence should be reworded for more clarity.
- The nomenclature used for figure sub-panels (e.g. Fig 2aiv) is confusing and uncommon.

Reviewer #3 (Remarks to the Author):

In their manuscript, Alberto et al use MEG in anaesthetized monkeys to map the network activation arising from optogenetic stimulation in frontal cortex and hippocampus. This work provides an important step forward for the development of optogenetic techniques in the primate brain and what one can expect from MEG measurements of optogenetic stimulation. While it therefore primarily is written as a methods paper, important technical details are lacking that prevent me from fully embracing the manuscript in its current form.

Major concerns:

1. It is not clear how the 750 μm^3 resolution claimed in the abstract is experimentally derived. As it seems to relate primarily to the beamformer method, it should be better explained, already from the abstract.
2. Can the authors be sure their result are indeed due to direct ChR2 stimulation? They should show details of spiking activity from the optrodes. The authors show data from a single session 2 weeks after injection (Figure 1C). This is fine, but it would be good to clarify that assessment during later time points uses the same parameters, including number of trials. From Figure 3b v., it seems responses to hippocampal stimulation have a latency of around 50 ms, suggestive of a network mediated effect. 'Activation' in Figure 2Bv., especially in the LFP, does not look very convincing and substantially smaller in amplitude compared to what is presented in Figure 1. Why is this the case?
3. Methods section: Clarify injection procedures, ie relation between coordinates and target structures, how many μl was injected in total per target structure? It would be good to know to the extent and sensitivity of the expression in the two target structures and whether retrograde labeling might have contributed to some of the network effects. A distance of 0.7 mm between electrode and optical fiber seems much, could you please explain the rationale? Line 391 and subsequent section on control for photoelectric effects: Could you please show plots confirming the absence of artifact in saline and in pre-expression control recordings. Which intensities of light are

used and how does the light distribute in space?

4. Figure 1ai and 1bi: It seems for this several methods have been combined, including fluorescence and MR imaging. The legend mentions a white line which I couldn't see. Or do the authors mean the large bright area in ai. that looks like an electrode? The legend also mentions colocalization of the electrode. But it is not clear to what. I presume this refers to colocalization between fluorescence and electrode in ai., but this is not entirely apparent, especially from a. ii/iii and even less clear for the situation in b.

5. For the data summarised in Figures 1d and e, it is not clear what these averages represent. Data from 1 or 2 or 3 monkeys, sessions, trials? Please provide the details for this figure in the legend, so one can fully understand and appreciate it. Most of the figures are MRI slices containing small yellow or green dots, for which the electrophysiological basis seems unclear, requiring the open-minded reader to believe in them.

6. Figure 4 and 5 show examples of network propagation. How reliable and consistent are these networks across the 3 monkeys used for the study. Can the authors say anything about the timing of the propagation?

Minor points:

7. Throughout the paper, different optical stimulation parameters are used (square pulses, sawtooth, sinusoidal), the authors do not show at all why such different parameters were used or if there are any differences due to the different parameters.

8. It is a good starting point of using MEG for deeper sources, however, the human brain is larger than that of an NHP; claiming similar structures can be detected in the human brain might be a bit of a stretch.

9. The figures should include some form of animal name/numbers to differentiate them. Right now, animals are referred to based on what other figures they appeared in which is confusing and harder to read.

10. A schematic for stimulation parameters (wave shape/frequency) should be displayed in the figures if they are different between the different subplots.

11. Line 78: how many trials/pulses do the evoked potentials figures represent?

12. Line 79: Please indicate that the values between brackets are for peak amplitude (mV).

13. Line 101: any reason for that specific number of trials and such stimulus presentation time?

14. Line 106/ Fig S1: with white flashes presented at the lower left visual field, maybe activation will be a bit more dorsal?

15. Line 126: explains pseudo t scores a bit talking about Fig 2aiv; the effects do not look as localised as the rest of the figures. Also, it might be better to explain pseudo t scores earlier when they are first mentioned.

16. Line 138/ Fig 2bi shows response to sine wave, any difference to square waves or the other higher frequency sine wave (line 139/40Hz)?

17. Line 144: Virtual electrodes are not really discussed except for two examples, and they are described as qualitatively similar to LFP evoked responses, maybe some sort of quantitative comparison would drive the point home that such a virtual electrode can be used instead of an actual one.

18. Line 156: why is the stimulation for one minute?

19. Line 165: why a single sawtooth pulse was used? Was it a train of pulses?

20. Line 166: typo capital F in figure

21. Figure 3bv: It might be hard to say anything about temporal aspects of responses due to a sawtooth pulse due to the gradual increase in power.

22. It might be clearer to include a time x-axis info in the LFP traces rather than just a scale bar.

Reviewer #4 (Remarks to the Author):

Alberto et al examine in this paper Magnetoencephalographical (MEG) measurements obtained within an experimental platform combining MEG-compatible optogenetic techniques in non-human primates (NHPs) to test the ability of MEG to detect signals in deep brain regions, such as the hippocampus. The authors also tested how neural activity propagates originating from arcuate sulcus and hippocampus. The main finding is the localization of optogenetically-evoked signals to known sources in the superficial arcuate sulcus of cortex and in CA3 of hippocampus. In addition, activation in subcortical and thalamic structures as well as extended temporal networks are detected.

This is a remarkable piece of research that uniquely combines optogenetics with MEG to provide causal evidence for the ability of MEG to detect activity in the hippocampus. In addition, the experimental platform has considerable potential for non-invasive neuroimaging. Accordingly, I feel that this paper would be suitable for publication provided that the authors address the following points:

- 1) As the authors will be aware, head movements are an important issue in source-localization, especially for deeper structures. Accordingly, more details are required on a) how head movements were monitored and b) how reliability of head-positions were assured across sessions
- 2) MEG-measurements were obtained during anaesthesia. The authors should address the potential impact of the anaesthetic regime used and how it could potentially influence the findings in this study.
- 3) I find the figure material (Figure 2-5, SI Figure 1) showing the localization of sources as well as their evolution across stimulation of limited value. Accordingly, it would be useful if the authors could come up with an alternative display approach.
- 4) The authors demonstrate ERFs in response to the visual stimulation (SI Figure 1). Curiously, the authors measured ERF responses in the arcuate sulcus and hippocampus. It would be useful if the authors perform more formal analyses that identify distinct peaks/ERFs to assess whether the stimulation resulted in reliable components consistent with the literature.

REVIEWER COMMENTS

Reviewer #1 (Remarks to the Author):

The authors aimed to develop an experimental platform combining MEG-compatible optogenetic techniques in non-human primates (NHPs) to test the ability of MEG/MSI to image deep signals. The study concludes that this is the first demonstration of accurate localization of deep sources within an intact brain using a novel combination of optogenetics with MEG/MSI. This is a very interesting study with clear methodological implications for future investigations. However, I have several major and minor concerns which I feel should be addressed via major revisions before this manuscript would be suitable for publication. One of the secondary goals is to provide evidence for the validity of using MEG to localize deep hippocampal sources. Please find my comments below:

Major:

1) Conceptual: There are conceptual aspects of the study that have not been clearly articulated for readers who are non-experts in the methods described in this manuscript, such as myself. I feel that the Introduction section does not provide a compelling rationale for conducting the study, nor what the study's implications might be going forward (other than those clinical in the field of epilepsy). For example, what is the significance of providing "a detailed investigation of the resolution capabilities" of MEG? What would be the implications of doing so?

We appreciate this critique and after careful review of the manuscript as initially presented, we agree with the concern. We hope that by conducting substantial restructuring and reframing of the Introduction, we have improved clarity and better motivated both the rationale and aims of the study. In doing so we hope to have additionally clarified the implications of the study moving forward. We hope that this concern has been addressed by near total revision of both the Introduction and Discussion sections of the manuscript.

For added background that is more historical than scientific, MEG installations are primarily driven by clinical use and much less commonly by research as MEG systems are expensive to install and maintain. This unfortunately limits its availability to the broader research community. It is commonly assumed that MEG cannot detect dipoles separated by less than 2 mm, nor that it can detect signals from below the cortical surface. As a result, MEG analysis techniques such as regularization and minimum norm estimation seek to boost the SNR of a signal by essentially averaging over a relatively large volume of tissue and/or restricting the signal to live on the cortical surface. Our data would suggest that such techniques would miss both fine-scale cortical activation patterns as well as subcortical generators, and that beamformers such as SAM are adequate to boost the SNR of deep signals. A more fine-grained analysis method could lead to better mapping of interictal spike generators in patients with epilepsy, in particular patients with mesial temporal or insular lobe epilepsy, as well as a detailed map of function in patients who are

candidates for brain tumor resection. MEG is underutilized for epilepsy surgery planning, and an increase in MEG utilization could lead to better outcomes for patients with epilepsy.

Additionally, nonhuman primates are a valuable translational tool through which the evolution of disease processes such as addiction is currently being investigated, and here we hope to provide the framework in which these animals may be imaged through MEG.

2) The paper does not flow well from rationale, to unknown or hypotheses to results and specific conclusions that address the unknown or hypothesis. It comes across as a “tour de force” in the combination of various techniques but is unclear why were they use in this specific way in the first place. For example, it is stated as important that it is possible to use human MEG to evaluate function in this particular animal but why is that important?

Upon reflection the authors wholeheartedly agree with this critique and see it as an extension of the first major concern above. We hope that the revisions have helped to focus the narrative, clarify rationale, and in turn improve the flow. Again significant changes to the structure of the Introduction and Discussion sections are intended to address this concern. We have included additional detailed background here for the reviewers, and hope that the revisions to the Discussion and Introduction will clarify these points in the manuscript. Particularly lines 238-244 address this.

In this study, one of our primary goals was to establish the methodology for imaging nonhuman primates with MEG. We are aware of only four other MEG studies in NHPs, and we would like to utilize these species as a translational model of disease processes in several additional and ongoing studies. One of the fundamental limitations of MEG is that magnetic field strength decays as the inverse cube of distance. Additionally, the bulk of the neuromagnetic signal is derived from EPSPs in pyramidal cortical macrocolumns. Given these factors, many studies map MEG activity to the cortical mantle only, and indeed regularize the sensor weights in the hope of improving SNR, although this concomitantly reduces spatial resolution. Furthermore, modeling studies have suggested that approximately 4 cm² of tissue must be activated to generate a field detectable by the MEG pickup coils. To compound the issue, the vervet brain is about the size of a large lime, which means that the cortical surface is much further from most of the MEG sensors than an adult human head would be and that the neural generators both cover a smaller surface area and arise from a thinner layer of grey matter.

One underappreciated point is that MEG sensors have different geometries depending on the helmet manufacturer, and axial gradiometers, which our system employs, have a much greater depth sensitivity than planar gradiometers, which tend to pick up primarily surface signals. A second factor is that SAM is not an inverse solution; it is a spatial filter. Inverse solutions such as the minimum norm estimator invert the sensor signals and constrain the signals such that they are

imposed to arise from the cortical surface only. We and others have shown that SAM selectively passes signal from each voxel while suppressing signals from all other sources and simultaneously removing noise. Here, because SAM is unconstrained and is initialized with volumetric lead fields, it simultaneously maps both the initial optogenetic response while also capturing downstream activity in known networks. This is a simple demonstration that surface-level inverse solutions might be inappropriate because they constrain signals that actually arise from deep, subcortical generators and project them on the cortical surface instead. Furthermore, nonhuman subcortical structures simply do not have a surface area of 4 cm^2 , and the localization of activity to subcortical areas suggests that a much smaller volume of tissue can give rise to a detectable MEG signal. Additionally, subcortical structures such as thalamus often have a laminar profile, which would facilitate the summation of similarly oriented biomagnetic fields. We note too that signal was localized to white matter, and we suspect that MEG via SAM can localize the passage of signals through white matter pathways, although this latter hypothesis will require additional testing. Finally, the canonical use for MEG is in epilepsy, and MEG is chronically underutilized for presurgical mapping. By showing that an adult MEG system can map a monkey, we hope that more babies and young children with epilepsy will be able to receive a MEG without the need for a smaller, pricey, and rare infant MEG helmet.

3) Analysis: There is not an adequate rationale given for some aspects of the analyses and thus leads this reader to doubt the validity of the study conclusions. Please make clear why the pseudo t-score was used. Was it planned that pseudo t-scores would be used, and if not, why do the authors think that there was increased noise? This same question goes for use of unthresholded images. Along these lines, the authors should add a limitations section.

We note that the pseudo t-score bears a bit of a confusing and unfortunate name. This measure is derived from the application of dual-state synthetic aperture magnetometry (SAM). In this case, biomagnetic signals during active-state trials are compared to signals in pre-stimulus, or baseline trials and are expressed as a ratio of power in a given frequency band and on a per-voxel basis. This ratio also inherently contains estimates of background (environmental) noise in both active and baseline conditions, and this noise (in addition to trial-to-trial signal fluctuations) can also be taken as a measure of variance. Thus, given the degrees of freedom (number of trials, number of voxels) the pseudo-t metric is computationally very similar to the t-statistic. This was first derived by Robinson and Vrba as a variant of their SAM beamformer (reference #2) and has been commonly applied to map task-related activity. We have attempted to add a more detailed description of this analysis technique to the methods section, and we have provided a brief set of additional supporting references. See specifically lines 427-435.

Local maxima or minima (peaks) in SAM maps have been shown to be significant neural sources/generators (refs 2,8), and SAM can uniquely separate up to $n-1$ neural sources, for n as the number of sensors. Peaks can exist at multiple scales/amplitudes (common particularly in

epilepsy) and should be thresholded individually. Activity above the full-width half maximum (FWHM) of the peak amplitude is generally considered to comprise an activated area, but the FWHM varies from peak to peak and thus one map may have many thresholds. In the case of our experiments, the SAM peaks we report are all above threshold and are significantly different from noise, but the advantage of not thresholding any images is that subtle patterns of activation and possible information flow can be discerned.

4) Presentation of results: The results should be presented in a more organized manner as opposed to long in-text descriptions. For example, the information described in Lines 178-212 could be supplied in a table. In addition, some of the results presented in-text and in figures are largely duplicable rather than complementary. One such instance includes portions of Lines 149-152 (“The blue vertical line indicates the time of stimulus onset, in this case a 10 ms square wave pulse. Both the LFP and virtual electrode exhibit a rapid rise and peak in activity following stimulus onset, and both qualitatively share similar features and time courses on a single trial basis.”), which are almost exactly the same as the figure description in Lines 656-657 for Figure 2 (“Blue vertical line indicates stimulus onset (10 ms pulses). Both the LFP (white) and virtual electrode (red) exhibit a rapid peak following stimulation and have similar features and time courses.”).

We have attempted to remove redundancies between lines 149 and 656. We have additionally reformatted the results for presentation in a table (Table 1). We hope that this results presentation approach will improve organization and clarity for the reader.

Minor:

1) Lines 79-81: What units do these CI's relate to?

Thank you – this has been fixed to include the unit, mV.

2) Lines 250-251: add "to" after "can be used".

Thank you- this has been fixed.

3) Line 254: But others have already used MEG to detect signals from deep brain structures, as noted in the Introduction section.

Thank you. We have attempted to clarify that this is both difficult and controversial, although depth pickup is augmented here by axial gradiometers. See line 55 for qualifying statement regarding deep signal detection as well as lines 264-271.

4) Line 361: Can the authors provide a reference for the conventional three-point fiducial locations?

Yes, we have added a reference for our particular MEG system.

In summary, material in this paper appears valuable as a technical contribution of possible interest for clinicians and basic science investigators. The relatively disorganized narrative reduces enthusiasm. The paper could gain from a more structured presentation format, focusing

around the resolution of a primary and perhaps secondary questions. The main take-home messages appears diluted by various claims that leave the read somewhat confused.

We thank you for these constructive comments and hope that the additions clarify some of the scientific and methodological problems we aim to address in this manuscript.

Reviewer #2 (Remarks to the Author):

Alberto and colleagues used viral vector-mediated optogenetics in anesthetized vervet monkeys to assess the ability of MEG-based techniques to detect changes in neural activity in the cortex and hippocampus. They found that optical stimulation of ChR2-expressing sites elicited MEG activity near the injection sites and in several downstream areas. Optical stimulation of transduced sites in the arcuate sulcus elicited activity in regions of the cortico-striatal-pallido-thalamocortical motor network, whereas optical stimulation of CA3 elicited activity in the temporal network. The authors conclude that this union of techniques (optogenetics and MEG/MSI) will be useful for exploring causal functional relationships between brain areas.

The manuscript is a valuable addition to the literature, although the novelty of the results should be articulated more clearly. Optogenetics has already been combined with fMRI in the macaque monkey (Ohayon et al., Journal of Neuroscience, 2013). Interestingly, in this previous study, optical stimulation of a cortical region (FEF) near the arcuate sulcus failed to elicit fMRI activity whereas electrical stimulation in combination with optical stimulation did. This relevant previous study is not cited or leveraged in the current manuscript. I believe that the authors are missing a unique opportunity to draw on similarities/differences between the two studies and between the resolution of MEG/fMRI to strengthen their claims of novelty.

We thank you for this observation and have attempted to address this point in the discussion section. We have also added this reference and a second one to facilitate the comparison between MEG and fMRI for detecting the optical response. See lines 215-219.

Aside from the above concern, there are other concerns related to methodology and interpretation that should be addressed. These points are outlined below.

Major concerns

- Several important details about the AAV vector injections are missing or unclear. What was the titer of this virus? And is there a reason the authors chose the AAV10 capsid serotype? Did the

authors inject only 4 microliters of vector at one site in each area? Did the authors deliver the vector at several depths in each area?

The AAV2/10 serotype was chosen because it is highly effective at transducing neurons in many species, including non-human primates. The virus was provided as a gift by Dr. Caroline Bass, and unfortunately no titer is available. Despite the titer being unknown, this vector has been very effective in our early rodent studies and prior to undertaking this study we demonstrated clear efficacy in NHPs in unpublished pilot data. See line 85 and lines 320-323.

- The histological images in Figure 1 are hard to evaluate. Is the green signal shown native fluorescence or was immunohistochemistry performed? Can the authors provide higher magnification images that clearly show transduced cell bodies in each area? And how long after the vector injection was this animal terminated? Finally, did the authors attempt to relate the spatial extent of vector transduction in this animal assessed histologically to the spatial extent of the MEG signal elicited in this animal?

The green signal is native EYFP expression, the figure legend has been updated to reflect this.

Due to high levels of ChR2-EYFP expression in the neuropil, it can be difficult to discern cell bodies at the injection site itself without oversaturating the image. However, single cells are clearly visible in areas of lower expression. An inset was included for hippocampus showing a transduced cell near the injection site. See Figure 1, inset.

The spatial extent of optogenetically activated cells is a function of both the transduction and the light spread from the fiber tip. Based on the parameters of the experiment we estimate a maximum of 1mm³ of activated tissue, corresponding to spatial resolution of analysis. See line 390-393.

- The authors use a variety of optical stimulation protocols (square, sine, saw-tooth) and different laser modulation frequencies. However, the authors never explain the rationale for these different stimulation protocols or their utility for performing functional mapping studies with MEG. It is currently unclear to the reader if/why the different stimulation protocols or wavelengths make a difference.

We apologize for the lack of clarity here. Originally, we were uncertain which types of stimulation would evoke a detectable MEG response so a variety of inputs were tried. The single square wave and ramp signals enable us to test whether brief inputs can be detected, and the ramp stimuli were thought to more gently induce the population response. The sine wave stimuli can mimic various aspects of ongoing resting state activity with the future goal of potentially enhancing or inhibiting endogenous brain activity and/or tracking the flow of information through brain circuits as a function of time/frequency content.

Ultimately in this manuscript we are able to demonstrate that magnetic source imaging with SAM is robust to stimulus waveform shape. See Lines 158-162.

- The supplementary figure shows a collection of brain regions engaged by visual stimulation through the eye. I'm confused about why this visual stimulus doesn't elicit activity in ALL visual areas (both cortical and subcortical). Is this a limitation of the MEG/MSI technique? Or is this an effect of anesthesia? Or are the authors only selecting to show a subset of the areas activated?

We apologize for the lack of clarity here. In addition to localizing activity to visual cortex, SAM also identified activity in the optic nerve, probable LGN (the location is consistent with known anatomy but the MRI is a bit fuzzy), and superior colliculus, as well as other brain regions associated with the visual network. We have added additional images of SAM visual peaks to Figure S1; these peaks were identified during the original beamforming process but had been omitted from the original submission for clarity.

- It would be useful to give readers a sense of how the magnitude of the LFP and/or MEG signal induced by visual stimulation compares to that induced by optical stimulation.

We have added the average visual event-related field to supplementary figure 1 to illustrate this. The peak to trough amplitude of the visual evoked field is about 30 nA-m, which is about the same as the amplitude of the arcuate source series (~45 nA-m) in response to 50 ms square wave light pulses. The square wave optical inputs are relatively modest in terms of impact; the sawtooth input to hippocampus elicited source series activations with an amplitude of about 1500 nA-m.

- Did the authors vary the intensity of optical stimulation and attempt to estimate the ability to detect neural activity with MEG? It could be useful to estimate the lowest light intensity that elicits MEG activity and how that might relate to the spatial resolution of the MEG signal.

Yes, this was performed as part of the battery of experiments; here we focus first on the basics of detecting optogenetic responses with MEG and plan to explore this further in subsequent studies and with more subjects.

- The section on "Functional mapping" reads like an exhaustive list of downstream areas where MEG activity is detected following optical stimulation near the injection site. What should the reader take away from this list? Are there areas that were expected to be connected to the arcuate sulcus and CA3 that do not show up in the MEG analyses?

MEG mapping is commonly constrained to the cortical surface only, in particular when alternative analyses to SAM such as the minimum norm estimator (MNE) are employed. SAM maps are best instantiated as volumetric, and not surface measurements; in particular SAM does not suffer from depth uncertainties that are seen in applications of the MNE. Initially, we wished to localize a MEG response to optogenetic stimulation, but here we also show the advantage of an unconstrained volumetric approach in that a broad temporal network is automatically uncovered in the same analytical process in which optogenetic activity in the hippocampus is mapped. We suspect that there is a frequency-dependent relationship for both hippocampal and arcuate stimulation and the identity of downstream structures that appear to be activated. We would like to explore this further by manipulating the time and frequency content of the optical stimulus. Similarly, while a biomagnetic response was elicited from arcuate cortex, downstream structures comprising the CSPTC network were also activated. This also demonstrates that a volumetric and unconstrained approach to MEG mapping provides additional benefits to cortical surface maps because whole functional networks, including subcortical peaks, can be revealed and their interdependencies later probed by examining their source series.

Minor concerns

- The authors use the words “transfected/transfection” incorrectly (Lines 72, 76, 93, 121, 178). “Transfection” typically refers to gene delivery via a plasmid not a virus. “Infection” refers to gene delivery via a virus, which is the case for this manuscript. However, it is more common in studies of viral vector-mediated optogenetics to use the words “transduced/transduction”, which are equivalent in meaning to “infection”. It would be best if the authors revisit their usage, perhaps referring to sites of interest as “transduced sites” or simply “injection sites”?

Thank you for this observation. We have replaced all instances of transfection with transduction in the manuscript.

- Figure 1: Panel C: It would be more appropriate to label these data based on the number of weeks post-injection rather than “pre-expression” and “post-expression”. We cannot be entirely sure that there is no expression at 2 weeks; we only know that there is no detectable activation at 2 weeks. Panels D and E: Are these data from a single session in one animal - or multiple sessions and animals?

Thank you we have addressed this imprecise language and clarified the figure legend for panels D and E to indicate that these data are from a single animal across multiple time points.

- As a non-expert in MEG analyses, it is unclear to me what the “virtual electrode” analysis entails and whether this is a widely used measure in the field. Relatedly, it is unclear to me whether the way that the authors represent SAM/MEG results in Figures 2–5 (single points with

weights) is common in the MEG literature. It might be useful to comment on these points to accommodate broader readership.

Virtual electrodes (source series) can be extracted from any magnetic source imaging technique. All MSI techniques weight the contributions of the MEG sensors to a hypothetical unit dipole at a particular location and orientation, and these weights can then be applied to the MEG channels to produce a readout of the dipole time series at that point in the brain. These virtual electrodes bear a very similar time frequency content to the signal obtained from an actual electrode and indeed look similar. We and others have shown clinically (reference 13) that MEG virtual electrodes produce similar results to implanted grids or strips in patients with epilepsy. There are subtle signal differences that have to do with impedance, tissue density, etc. in the signal readout between implanted electrodes and the virtual electrodes (which are not sensitive to these effects) but basically both measure the local field potential in a particular region. We have attempted to provide a bit more background on SAM and its outputs in the Methods section. See lines 422-436 specifically and the Analysis section of Online Methods for general discussion.

- Line 54: Not all opsins are light sensitive “ion channels”—some are pumps.

Thank you- we have fixed this. See line 61.

- Line 435: I think there is a typo here. Do the authors mean “in vivo” not “in vitro”? And how does one test the optrode in post mortem tissue? This sentence should be reworded for more clarity.

We have removed this line; we thank the reviewer for noticing it and it is from an earlier draft. The more complete statement is found in Online Methods, Controlling for Photoelectric Effects, and we have added a supplementary figure to address this. We performed postmortem MEG recordings to assess whether the detected biomagnetic signal was a result of the photoelectric effect but found that there were no differences between the non-stimulated condition, hippocampal nor arcuate stimulation, nor MEG signals from an empty room.

- The nomenclature used for figure sub-panels (e.g. Fig 2aiv) is confusing and uncommon.

We have used this format to be consistent with Nature formatting guidelines. We hope that by re-organizing the presentation of the results into a table that the references to figure sub-panels will be clearer and less confusing.

Reviewer #3 (Remarks to the Author):

In their manuscript, Alberto et al use MEG in anaesthetized monkeys to map the network activation arising from optogenetic stimulation in frontal cortex and hippocampus. This work provides an important step forward for the development of optogenetic techniques in the primate brain and what one can expect from MEG measurements of optogenetic stimulation. While it therefore primarily is written as a methods paper, important technical details are lacking that prevent me from fully embracing the manuscript in its current form.

Major concerns:

1. It is not clear how the 750 μm^3 resolution claimed in the abstract is experimentally derived. As it seems to relate primarily to the beamformer method, it should be better explained, already from the abstract.

We apologize for the lack of detail here. Essentially, the lead field solution is initialized with a grid of unit dipoles uniformly spaced with a distance of 750 μm and the beamformer weights are applied to the lead field to produce the SAM map. We found that with a larger spacing, such as 1 or 2 mm, that the optogenetic signal was blurry and unable to be adequately resolved. We have attempted to address this omission by adding a supplementary figure to show how changing voxel size affects the localization of the optogenetic stimulus.

Please see lines 438-448 and supplemental figure 2.

2. Can the authors be sure their result are indeed due to direct ChR2 stimulation? They should show details of spiking activity from the optrodes. The authors show data from a single session 2 weeks after injection (Figure 1C). This is fine, but it would be good to clarify that assessment during later time points uses the same parameters, including number of trials. From Figure 3b v., it seems responses to hippocampal stimulation have a latency of around 50 ms, suggestive of a network mediated effect. 'Activation' in Figure 2Bv., especially in the LFP, does not look very convincing and substantially smaller in amplitude compared to what is presented in Figure 1. Why is this the case?

Yes, it would be likely that there is somewhat of a network effect in both the hippocampal and cortical stimulation. Activity spreads beyond the initial site of stimulation and may elicit a greater response outside the optogenetic zone depending on the nature of the time and frequency content of the stimulus. We hypothesize this may have to do with a combination of the input signal and the resting state frequencies exhibited by each structure. Both the LFPs and source series presented in figures 2 and 3 are for single trial responses and are unaveraged, whereas the responses in figure 1 are averaged. The electrophysiological recording methods for figure 1 differs from that in figures 2 and 3 as well; in the first case the LFPs were obtained from a dedicated electrophysiology rig capable of recording single units; in the second two cases the LFPs were recorded via the EEG subsystem of the MEG itself. The EEG system was not

intended for invasive recordings, but in this case it permits simultaneous acquisition with the MEG. Owing to the difficulty of running the invasive recordings through the EEG subsystem not all of the experiments had simultaneous MEG/LFP recordings.

3. Methods section: Clarify injection procedures, ie relation between coordinates and target structures, how many ul was injected in total per target structure? It would be good to know to the extent and sensitivity of the expression in the two target structures and whether retrograde labeling might have contributed to some of the network effects. A distance of 0.7 mm between electrode and optical fiber seems much, could you please explain the rationale? Line 391 and subsequent section on control for photoelectric effects: Could you please show plots confirming the absence of artifact in saline and in pre-expression control recordings. Which intensities of light are used and how does the light distribute in space?

Thank you, this information is now included in the Online Methods under “surgery” and has been edited to improve clarity. We did not observe any indication of retrograde labelling but it cannot be ruled out and it is certainly possible that activation of retrograde labelling of afferent terminals contributed to the observed network activity. We think this possibility is unlikely however given the lack of retrograde label observed with AAV2/10 serotype in different model organisms as compared to other serotypes, eg AAV5.

The distance of 0.7mm was chosen primarily so that the fiber was positioned well away from expressing cells in order to avoid damage to the target structure. The fiber diameter, 400 μm , allowed for sufficient light for activation.

Pre-expression control recordings are shown in figure 1C. A plot showing the absence of prior to expression and in post-mortem recordings has been added as supplemental figure 4.

The light intensities used are included in the methods section under “Electrophysiology Recordings” and a reference to previous estimates of light spread from implanted fibers has been added. See line 392.

4. Figure 1ai and 1bi: It seems for this several methods have been combined, including fluorescence and MR imaging. The legend mentions a white line which I couldn't see. Or do the authors mean the large bright area in ai. that looks like an electrode? The legend also mentions colocalization of the electrode. But it is not clear to what. I presume this refers to colocalization between fluorescence and electrode in ai., but this is not entirely apparent, especially from a. ii/iii and even less clear for the situation in b.

Thank you for drawing this to our attention. We have fixed this oversight.

5. For the data summarised in Figures 1d and e, it is not clear what these averages represent. Data from 1 or 2 or 3 monkeys, sessions, trials? Please provide the details for this figure in the legend, so one can fully understand and appreciate it. Most of the figures are MRI slices containing small yellow or green dots, for which the electrophysiological basis seems unclear, requiring the open-minded reader to believe in them.

Figure 1d and e are averages of multiple trials from a single subject. We have clarified this in the legend. Thank you.

We have attempted to add more of a description of SAM in the methods section. Basically, only local maxima or minima in the SAM map are significant, and in this case the SAM peaks localized to the location of optical stimulation as well as downstream structures known to be involved in relevant networks, or in the case of visual stimulation, to structures within the visual network. We have added a supplementary figure to show the effects of changing voxel size on localization, and these maps also illustrate the magnetic field topography, the event-related synchronization and desynchronization, and the narrow distribution of the fields around the site of arcuate stimulation. In addition to the whole brain maps presented in figures 2 and 3, we hope that this supplementary figure better illustrates the MEG imaging process.

6. Figure 4 and 5 show examples of network propagation. How reliable and consistent are these networks across the 3 monkeys used for the study. Can the authors say anything about the timing of the propagation?

In addition to the animals presented in this manuscript, we have instrumented a fourth subject for a complementary study. To date, stimulating in both hippocampus and in arcuate cortex appears to cause propagation through the temporal and CSPTC networks in multiple subjects, respectively. The sine wave stimuli appear to cause greater propagation than the brief square wave inputs, and this could be related to the time-frequency parameters of the inputs. We are conducting network analyses to quantify these relationships, although this is a work in progress.

Minor points:

7. Throughout the paper, different optical stimulation parameters are used (square pulses, sawtooth, sinusoidal), the authors do not show at all why such different parameters were used or if there are any differences due to the different parameters.

We have attempted to address the utilization of different stimulus inputs at the beginning of the results section. See lines 158-162.

Originally, we weren't sure what sorts of stimuli would be detectable by MEG and were uncertain whether very brief signals such as 10 ms pulses could be recovered. However, we found that all inputs could be mapped; the chief difference between the inputs is that the MEG responses tend to mirror the time-frequency aspects of the inputs such that brief inputs cause brief brain responses and that longer sine wave inputs engage longer brain responses that tend to resonate at about the same frequencies as the sines. This is reflected in the SAM beamforming parameters.

8. *It is a good starting point of using MEG for deeper sources, however, the human brain is larger than that of an NHP; claiming similar structures can be detected in the human brain might be a bit of a stretch.*

The volumes are certainly different! Part of the problem is that the monkey brain is so small, and the head surface is very distant from most of the MEG sensors, and much further than that of a human. Additionally, the volume of tissue for subcortical brain structures is much smaller than those in humans, and such structures should thus generate much weaker fields. Despite this, we can detect MEG signals both cortically and subcortically in the NHP preparation. We have published on detecting hippocampal signals from patients with epilepsy, and the localizations were subsequently confirmed by invasive monitoring as part of the surgical process. The NHP recordings are a second piece of evidence to suggest that subcortical brain regions are accessible to MEG.

9. *The figures should include some form of animal name/numbers to differentiate them. Right now, animals are referred to based on what other figures they appeared in which is confusing and harder to read.*

Thank you. We have provided animal ID numbers to disambiguate the subjects. See table 1.

10. *A schematic for stimulation parameters (wave shape/frequency) should be displayed in the figures if they are different between the different subplots.*

We have attempted to provide clarity by referring to the stimulus type in an organized fashion in Table 1. See also lines 158-162 for discussion of different stimulus types.

11. *Line 78: how many trials/pulses do the evoked potentials figures represent?*

We have added this to the manuscript at the relevant figures as well as added this information to a newly included table (Table 1).

12. *Line 79: Please indicate that the values between brackets are for peak amplitude (mV).*

Thank you, this has been fixed.

13. *Line 101: any reason for that specific number of trials and such stimulus presentation time?*

Well, one of the light pulses was not synchronized with the MEG data recording, so the number was reduced from the original 120 to a final of 119. The light pulse was brief because we wanted to see whether a brief and moderately intense signal could be mapped by SAM. The interstimulus interval was selected at 6 seconds to avoid overlapping effects of stimulation, while still maintaining a reasonably short duration of anesthesia for the animal.

14. Line 106/ Fig S1: with white flashes presented at the lower left visual field, maybe activation will be a bit more dorsal?

We have added more of the visual network to this figure to better illustrate what was detected by the beamforming.

15. Line 126: explains pseudo t scores a bit talking about Fig 2aiv; the effects do not look as localised as the rest of the figures. Also, it might be better to explain pseudo t scores earlier when they are first mentioned.

We have attempted to provide more details on the beamforming process in the Methods section. We suspect that the volume of tissue activated is a function of the optical stimulation type, and we are working to explore this more with additional subjects in a subsequent and ongoing study.

16. Line 138/ Fig 2bi shows response to sine wave, any difference to square waves or the other higher frequency sine wave (line 139/40Hz)?

Yes, the beamforming indicates that the biomagnetic signal tracks the input stimulus time/frequency parameters. We are conducting a network study to quantify the effects of these different parameters on information flow.

17. Line 144: Virtual electrodes are not really discussed except for two examples, and they are described as qualitatively similar to LFP evoked responses, maybe some sort of quantitative comparison would drive the point home that such a virtual electrode can be used instead of an actual one.

The virtual electrodes are a very nice feature of MEG analysis, and there is some suggestion in the literature that the virtual electrodes may augment, reduce, or possibly replace electrocorticography in patients with epilepsy. Our goal here was to show that the MEG virtual electrodes had a similar temporal profile to the simultaneously recorded LFP, and in particular to show that the signal did not degrade nor substantially differ in the hippocampus. There are slight differences noted in the literature between the two types of signals owing to the resistivity of brain tissue to the electrical current flow, whereas the magnetic fields pass through bodily tissues without attenuation.

18. Line 156: why is the stimulation for one minute?

We were unsure how long the signal needed to be to generate a detectable MEG response, but we subsequently found that brief signals were recoverable.

19. *Line 165: why a single sawtooth pulse was used? Was it a train of pulses?*

The sawtooth signal was a single waveform that was repeated on multiple trials. We have changed the language of the manuscript to the more accurate “ramp” nomenclature.

We were initially unsure as to what signals could be detected by MEG, but so far the different sorts of inputs have had a measurable MEG output. See lines 160-164.

20. *Line 166: typo capital F in figure*

Thank you.

21. *Figure 3bv: It might be hard to say anything about temporal aspects of responses due to a sawtooth pulse due to the gradual increase in power.*

That is certainly true. The goal here is to show that the LFP and the MEG virtual electrodes have a similar evolution.

22. *It might be clearer to include a time x-axis info in the LFP traces rather than just a scale bar.*
We have added a time axis to the LFP traces.

Reviewer #4 (Remarks to the Author):

Alberto et al examine in this paper Magnetoencephalographical (MEG) measurements obtained within an experimental platform combining MEG-compatible optogenetic techniques in non-human primates (NHPs) to test the ability of MEG to detect signals in deep brain regions, such as the hippocampus. The authors also tested how neural activity propagates originating from arcuate sulcus and hippocampus. The main finding is the localization of optogenetically-evoked signals to known sources in the superficial arcuate sulcus of cortex and in CA3 of hippocampus. In addition, activation in subcortical and thalamic structures as well as extended temporal networks are detected.

This is a remarkable piece of research that uniquely combines optogenetics with MEG to provide causal evidence for the ability of MEG to detect activity in the hippocampus. In addition, the experimental platform has considerable potential for non-invasive neuroimaging. Accordingly, I feel that this paper would be suitable for publication provided that the authors address the following points:

1) As the authors will be aware, head movements are an important issue in source-localization, especially for deeper structures. Accordingly, more details are required on a) how head movements were monitored and b) how reliability of head-positions were assured across sessions

Yes, head motion will definitely interfere with signal localization. We have utilized head localization during each scan, and because of the anesthesia, the head motion is minimal (< 200 µm). The animals are also tattooed at the nasion and preauricular points to facilitate consistent fiducial placement from scan to scan. See lines 414-415.

2) MEG-measurements were obtained during anaesthesia. The authors should address the potential impact of the anaesthetic regime used and how it could potentially influence the findings in this study.

We have experimented a bit with anesthesia and the NHP preparation, although that is the subject of a different manuscript. In rodents, gaseous anesthetics largely ameliorate single unit cortical responses, at least in our experience. Propofol is something that we have used clinically for our patients with epilepsy as it seems to preserve cortical responses, and in a separate series of tests we have found that NHP auditory pathways are intact under propofol. We might expect that anesthesia would reduce cortical excitability and/or signal propagation, but both the optogenetic and visual stimuli produced network effects that seemed to respect known circuitry. Line 204 touches on the ability of the preparation to overcome anesthesia effects.

Anesthesia is certainly a limitation for this study and we are currently developing MEG recording in fully conscious monkeys.

3) I find the figure material (Figure 2-5, SI Figure 1) showing the localization of sources as well as their evolution across stimulation of limited value. Accordingly, it would be useful if the authors could come up with an alternative display approach.

We have now included these results as part of a table which we hope will improve usefulness and clarity (Table 1).

4) The authors demonstrate ERFs in response to the visual stimulation (SI Figure 1). Curiously, the authors measured ERF responses in the arcuate sulcus and hippocampus. It would be useful if the authors perform more formal analyses that identify distinct peaks/ERFs to assess whether the stimulation resulted in reliable components consistent with the literature.

This is an interesting point. We have extracted the averaged visual evoked field for one of the V1 peaks in supplementary figure 1 and it has components that are comparable to those for humans in another MEG study (Hagler et al., 2009). We added that to the figure, although regrettably we do not have indwelling electrodes in V1 against which to compare the signal.

** See Nature Research's author and referees' website at www.nature.com/authors for information about policies, services and author benefits.

Our flexible approach during the COVID-19 pandemic

If you need more time at any stage of the peer-review process, please do let us know. While our systems will continue to remind you of the original timelines, we aim to be as flexible as possible during the current pandemic.

This email has been sent through the Springer Nature Tracking System NY-610A-NPG&MTS

Confidentiality Statement:

This e-mail is confidential and subject to copyright. Any unauthorised use or disclosure of its contents is prohibited. If you have received this email in error please notify our Manuscript Tracking System Helpdesk team at <http://platformsupport.nature.com> .

Details of the confidentiality and pre-publicity policy may be found here <http://www.nature.com/authors/policies/confidentiality.html>

REVIEWERS' COMMENTS

Reviewer #1 (Remarks to the Author):

My concerns have been addressed

Reviewer #2 (Remarks to the Author):

Alberto and colleagues have revised the manuscript substantially. They have addressed the major/minor concerns I raised previously except for one, which is outlined below. I am confident that the authors can include these additional methodological details in the final version.

Given that this is one of few monkey optogenetics manuscripts, the details of vector injection should be clarified further. Did the authors inject only 4 microliters of vector at each "coordinate" or 4 microliters in total per animal? How many "coordinates" were injected per area and/or per animal? Did the authors deliver the vector at several depths at each "coordinate" or only at one depth? These injection details are fairly standard in manuscripts that employ monkey optogenetics and will help guide subsequent future studies.

Reviewer #3 (Remarks to the Author):

Most of my previous comments have been addressed. There are three outstanding aspects that I think should be improved.

- The first point is about convincing readers further the observed effects are indeed due to optogenetic stimulation. The authors have chosen a longitudinal approach showing that stimulation became only effective after successful expression. I was hoping that the authors could show successful neural spiking activity from the stimulation site demonstrating indeed direct suprathreshold optogenetically elicited activation. However this seems not feasible. I think rather than brushing over this aspect, I believe the authors should be explicit about this aspect and perhaps consider including a limitation statement. A related aspect here is also the consideration whether heating might have played a role in generating their effects. This is discussed in Stujenske et al (10.1016/j.celrep.2015.06.036).
- I see that the authors discuss Ohayon et al (<https://doi.org/10.1523/JNEUROSCI.2675-13.201>) following the advice of a co-reviewer. However I believe that the method description is inaccurate, as the measured fMRI relied on an external contrast agent and not BOLD. The authors should then also cite the work of Gerits et al (10.1016/j.cub.2012.07.023) who used pretty much the same experimental approach but did find optogenetically induced activation. For readers of the three studies it would be also interesting to hear why different results were obtained with fMRI vs MEG. As far as I understand the two fMRI studies reported not any activation of the basal ganglia circuit, but focused on cortical activations that were not apparent in MEG.
- Injection methods. Can you please clarify whether you injected only at one location per brain structure. For example, did you inject 4 uL of virus into the Hippocampus and another 4 uL of virus into FEF? Or is the total injected volume larger? It would be very interesting to know this information as many labs are using much larger volumes that they inject at several locations per brain structure.

Reviewer #4 (Remarks to the Author):

The authors addressed my comments.

Reviewer #1 (Remarks to the Author):

My concerns have been addressed

Thank you.

Reviewer #2 (Remarks to the Author):

Alberto and colleagues have revised the manuscript substantially. They have addressed the major/minor concerns I raised previously except for one, which is outlined below. I am confident that the authors can include these additional methodological details in the final version.

Thank you.

Given that this is one of few monkey optogenetics manuscripts, the details of vector injection should be clarified further. Did the authors inject only 4 microliters of vector at each “coordinate” or 4 microliters in total per animal? How many “coordinates” were injected per area and/or per animal? Did the authors deliver the vector at several depths at each “coordinate” or only at one depth? These injection details are fairly standard in manuscripts that employ monkey optogenetics and will help guide subsequent future studies.

Thank you for bringing this lack of clarity to our attention. We have revised the Methods section at line 330-331 to provide further detail. Briefly, each target location was injected one time with 4 microliters of vector; the injection needle was not moved during the injection. A total of 8 microliters of vector was delivered per animal divided equally between two sites.

Reviewer #3 (Remarks to the Author):

Most of my previous comments have been addressed. There are three outstanding aspects that I think should be improved.

- The first point is about convincing readers further the observed effects are indeed due to optogenetic stimulation. The authors have chosen a longitudinal approach showing that stimulation became only effective after successful expression. I was hoping that the authors could show successful neural spiking activity from the stimulation site demonstrating indeed direct suprathreshold optogenetically elicited activation. However this seems not feasible. I think rather than brushing over this aspect, I believe the authors should be explicit about this aspect and perhaps consider including a limitation statement. A related aspect here is also the consideration whether heating might have played a role in generating their effects. This is discussed in Stujenske et al (10.1016/j.celrep.2015.06.036).*

Optogenetic stimulation produces synchronous activation of many neurons, producing a poly-spike which does not allow for waveform-based spike sorting and low-impedance electrodes such as those employed in our study are not suited to detect single units, although that would be an interesting

approach for a future study. However, optogenetically evoked field potentials are a well-established measure of population activity and are sufficient to clearly and unambiguously demonstrate an optogenetic response in our preparation. Please see Cardin et al. (10.1038/nature08002), Klorig et al. (10.1523/ENEURO.0229-18.201), and Buzsaki et al. (10.1038/nrn3241) for additional discussion on the nature of optogenetically evoked signals.

All of our stimuli were designed to be too brief and infrequent (ie. most reported experiments used 10 ms pulses w/ 6 s ISI, 0.167% duty cycle) for light-induced heating to be relevant according to the model developed in Stujenske et al. Even in the longest periods of stimulation reported we do not expect a significant contribution from light induced heating based on Stujenske et al.'s model as a 4 ms pulse width at 20 Hz corresponds to an 8.0% duty cycle. A reference to Stujenske et al has been added to the manuscript and a discussion of heating was added at lines 410-413. For additional clarity regarding optical stimulation parameters we added columns for light intensity, pulse duration, train duration, duty cycle, and fiber diameter to Table 1.

In reviewing the relevant sections a typographical error made it appear that optical stimulus intensity was on the order of watts (W), this has been corrected to milliwatts (mW). For added clarity we removed the reported command voltages as this is not easily translated to intensity unless one were to use our exact equipment.

• *I see that the authors discuss Ohayon et al (<https://doi.org/10.1523/JNEUROSCI.2675-13.201>) following the advice of a co-reviewer. However I believe that the method description is inaccurate, as the measured fMRI relied on an external contrast agent and not BOLD. The authors should then also cite the work of Gerits et al (10.1016/j.cub.2012.07.023) who used pretty much the same experimental approach but did find optogenetically induced activation. For readers of the three studies it would be also interesting to hear why different results were obtained with fMRI vs MEG. As far as I understand the two fMRI studies reported not any activation of the basal ganglia circuit, but focused on cortical activations that were not apparent in MEG.*

Thank you for raising this concern. After a careful re-reading of Ohayon et al., we suggest that the contrast mentioned by the reviewer is used only to improve analysis of the BOLD signal, but fundamentally the fMRI analysis conducted in Ohayon is based on the BOLD signal. The use of the BOLD signal for functional analysis is clear from the discussion where Ohayon et al. details the differences in their findings compared to Gerits et al. We note that Ohayon et al. includes a thoughtful discussion of Gerits et al. in which they conclude that the saccade-mediated task from Gerits et al. confounds the interpretation of the BOLD signal detected in the Gerits et al. study.

We agree with the Ohayon et al. interpretation of Gerits et al., but we can only speculate as to differences between fMRI and MEG observations. When considering the identification of basal ganglia, the methods detailed in Gerits et al. and Ohayon et al. make it difficult to determine the extent of their analysis, but it is clear that they are doing ROI-based analysis using a linear model. There are a number of points of deviation when comparing the fMRI analyses in Ohayon et al and Gerits et al to the MEG analysis conducted here: 1. Considerably fewer data points are used to analyze fMRI data (ie MEG data were sampled at 2400Hz); 2. Larger voxel sizes were used in fMRI analyses (1mm³ vs 0.75mm³); 3. No *a priori* location information was used in MEG analysis (i.e., this is not ROI based analysis); 4. SAM is not fit

to a “predicted” signal. Points 1 and 2 taken together make it such that fast, transient, highly-localized activity could be missed by BOLD signal analysis; considerable attention is paid to this point in the manuscript Discussion (Lines 221-236). Points 3 and 4 demonstrate that the SAM analysis conducted in this manuscript is unbiased. Potential sources of bias in fMRI analysis (e.g., ROI identification and model fitting) could explain additional differences between fMRI and MEG results. This is intrinsic to the types of data acquired and the way they are analyzed. The nature of MEG data and MEG data analysis is discussed in the Introduction (lines 31-43 and 72-80) and in the Methods section, subheading “Analysis” particularly lines 418-441.

We should also note that MEG data is directly related to neural activity whereas fMRI data is a surrogate signal thought to be correlated to neuronal activity. It is possible that some neural activity does not always have a corresponding BOLD signal change whereas any activity with coincident current will have an associated magnetic field. Whether or not weak magnetic fields reach the detector is another question. This study adds support that such weak signals are detectable even in high noise recording environments.

• *Injection methods. Can you please clarify whether you injected only at one location per brain structure. For example, did you inject 4 uL of virus into the Hippocampus and another 4 uL of virus into FEF? Or is the total injected volume larger? It would be very interesting to know this information as many labs are using much larger volumes that they inject at several locations per brain structure.*

We have revised the Methods section line 330-331 to provide further detail. Briefly, each target location was injected one time with 4 microliters of vector; the injection needle was not moved during the injection. A total of 8 microliters of vector was delivered per animal divided equally between two sites.

Reviewer #4 (Remarks to the Author):

The authors addressed my comments.